# Bibliometrics Evaluation of Scientific Journals and Country Research Output of Dental Research in Latin America Using Scimago Journal and Country Rank

**Gustavo Vaccaro** [1,2,3], **Pablo Sánchez-Núñez** [4,*] **and Patricia Witt-Rodríguez** [5]

1   Centre for Applied Social Research (CISA), Universidad de Málaga, 29010 Málaga, Spain
2   Institute of Biomedical Research of Malaga (IBIMA), 29590 Málaga, Spain
3   Secretary of Higher Education, Science, Technology, and Innovation (Senescyt), Guayaquil 090512, Ecuador
4   Department of Audiovisual Communication and Advertising, Faculty of Communication Sciences, Universidad de Málaga, 29010 Málaga, Spain
5   Faculty of Dentistry, Universidad de Guayaquil, Guayaquil 090512, Ecuador
*   Correspondence: psancheznunez@uma.es

**Abstract:** Innovations in dental sciences are potentially disruptive; however, the language barrier in the case of Latin America and the Caribbean (LAC) limits access to scientific studies. There is a necessity to measure the development of dental research across the LAC region, where economic power and postgraduate education access vary significantly. This article aims to analyze documents, citations, and journals and compare the SJR, H-Index, citation rates, and Co-occurrence Networks (Keywords) between dental journals published in LAC and the rest of the world, according to the report of Scimago Journal and Country Rank, between the years 1996 and 2020. Results show that Brazil leads dental research in the LAC, scoring the highest number of published documents, citations, and SJR metrics. The mean H-index and SJR of LAC dentistry journals are significantly lower than those of other regions ($p < 0.03$); however, there are no significant differences in the mean total citations in the last 3 years between LAC and other regions ($p > 0.15$). This suggests that the articles published in dentistry journals from LAC are being cited in similar proportions to the journals of other regions, but a large portion of these citations came from publications with low scientific impact.

**Keywords:** dental research; Latin America and the Caribbean; informetrics; scientometrics; Bibliometrics; research evaluation; SJR; Scopus

## 1. Introduction

The interest in dental research in Latin America and the Caribbean (LAC), as a biomedical field, has increased in the past decades [1]. Many new research topics are being constantly introduced, resulting in a broadening of the span of research conducted and produced [1,2]. Therefore, it is very challenging for dentists to keep up to date with the latest evidence because of the current large volume of dental research productivity [3]. There have been various potentially disruptive innovations in dental sciences in the last decade, such as dental robotics [4], stem cell treatment [5], and 3D printing prosthetics [6], among others, that could permanently change the way dental care is provided and modify dental treatment models. The adoption of new technologies in dental care is influenced by information acquired from professional journals, continuing education, industry marketing activities, and interactions with colleagues [7].

In the case of LAC, the inclusion of new technologies in everyday dental care is limited by factors such as the cost of new equipment, the availability of the technology, and specialized training. Moreover, oral diseases remain largely untreated because the treatment costs exceed the available resources in large portions of the LAC region [8]. Therefore, there is a possibility that the low priority of dental care in LAC might discourage dental research in the region.

There is a necessity to measure the development of dental research, including a variety of innovation types and populations across the LAC region, where the economic power and postgraduate education access vary significantly depending on the country and even within the same country. The scientific production within the context of universities is performed mainly by professors and research groups, with little participation from students [9]. Previous research suggests that dentistry professionals and the academic community in LAC are, in general, not attracted to publishing or participating in scientific production. This hinders innovation at a core level, and much research can go unpublished and therefore unnoticed by the scientific community. It is important to consider that innovation emerging from scientific research related to higher education can be translated to the productive sector [9–12]. In this regard, incorporating scientific evidence into dental practice requires that professionals apply the best available knowledge in the process of clinical decision-making. However, the language barrier in the case of LAC limits health care providers' access to scientific studies in order to find such evidence [13,14].

There are many professional incentive mechanisms in LAC that encourage scientists to publish original research in peer-reviewed high-impact journals [12]. Although there are regional cooperative bibliographical information systems and public electronic databases with numerous academic journals in the Spanish language, most of the high-impact dentistry journals are published in the English language [15]. This situation, in combination with poor instruction in article writing and scientific methodology, explains the low scientific production in biomedical fields in LAC [9,16].

Additionally, access to the outstanding scientific literature for the analysis of evidence-based dentistry is limited due to the restricted funding, underfunding, or non-existent funding in universities and higher research centers [17]. In many cases, states do not contribute to the payment of subscriptions to the main sources of information and international databases such as the Web of Science or Scopus [18,19].

Within this context, scientific journals are one of the main diffusion channels for scientific research, for private and public entities alike. Journals allow the advancements in a given field to be known internationally, benefiting the scientific community, and driving innovation, leading to an improvement in the quality of life [20].

The available methods to monitor the impact of the dissemination of research results as published scientific articles are peer review and bibliometric analysis [21]. The evaluation of scientific production is an indicator of the scientific and technological development within a community, nations, and professions [9]. The peer-review process is a crucial component in the publication of scientific articles. Highly cited dental journals use a peer review process to select articles that will be published, following strategies to ensure the quality and suitability of the manuscript for publication [22].

Bibliometrics is a quantitative analysis of published data. This analytical approach is used as a tool in research performance evaluation to help the identification of research needs, resource allocation, and decision-making support [21]. Bibliometric studies apply mathematics and statistics to quantitatively evaluate the scientific literature to highlight publishing trends in a scientific field and evaluate the impact of journals, articles, and researchers [23].

Bibliometrics has been used previously to critically evaluate the contribution of academics from LAC to the development of dentistry in the region by describing the characteristics of their publications, main research lines, sponsor institutions, and the kind of journals where these studies were published [15,24,25]. A recent study published by Cámara and Flores in 2021 that assessed the quality of dental research in Mexico concluded that there is low scientific productivity and a predominance of papers focused on researching or documenting already-studied problems and case reports from private dental practices involving well-known treatment procedures, with poor scientific contributions [10].

Many variables can be used to evaluate a specialized scientific journal, where two main factors are considered: the citation rates and the H-index [26]. The impact factor is an indicator that allows the assessment of the quality of a journal related to a knowledge area; it

measures the number of citations related to the number of articles published within a period. Similarly, the Scimago Journal Rank (SJR) is an indicator that assesses the impact of scientific journals that considers the citation networks to quantify the average prestige per article and can be used for journal comparisons [27]. Moreover, the H-index is a numeric indicator that also assesses the scientific production related to the number of citations received, defined as the largest value of H such that the journal has published at least H papers cited at least H times [28]. It has been debated whether the number of citations received by an article reflects its actual influence in the literature or whether it is a fair appraisal, since it does not represent the whole complexity of the research work [29]. Nevertheless, citation rates remain widely used as indicators of the influence of papers and journals in science [23]. There are further approaches to assessing the conceptual structure of research such as the analysis of the most frequently used keywords and their relationships between publications and authors through a map of the Co-occurrence Network. The Co-occurrence Network methodology is a powerful tool to find the relationships between social actors in research and identify patterns trends and preferences of interactions between researchers, groups, and organizations [30].

The citation process in scientific journals has links with intellectual heritage and is crucial for the incremental process of science [31]. In the case of LAC, the impact metrics of scientific journals specialized in dentistry and oral health can provide an overview of the current state of dental research in the region and provide research supporting agencies with valuable information to evaluate budgetary spending and distribute financial resources.

This research aims to analyze documents, citations, and journals and compare the SJR, H-Index, and citation rates between dental journals published in LAC and the rest of the world according to the report of Scimago Journal and Country Rank, between the years 1996 and 2020. This study aims to answer the following research questions (RQ):

- RQ1. Which are the countries in LAC with the highest number of publications and citations in dentistry?
- RQ2. Are there differences in the metrics associated with the publication output between LAC and other regions?
- RQ3. How are the most frequently used keywords related to dental research distributed?

In this regard, the null hypotheses tested in this research are as follows:

- $H1_0$. There are no differences in the mean H-Index of dentistry journals between LAC and Other Regions.
- $H2_0$. There are no differences in the mean SJR of dentistry journals between LAC and Other Regions.
- $H3_0$. There are no differences in the mean number of citations per document in dentistry journals between LAC and other regions.
- $H4_0$. There are no differences in the mean number of total citations of dentistry journals between LAC and other regions.

To do so, the paper is set out as follows: Section 2, defines the materials, methods, and the approach used in the analysis; Section 3 presents the results of our analysis; and, finally, Section 4 concludes with the discussion, conclusion, and upcoming investigation approaches.

## 2. Materials and Methods

### 2.1. Database and Data Acquisition

Scimago Journal & Country Rank was used as a source of information. The Scimago Journal & Country Rank is a publicly available portal that includes the journals and country scientific indicators developed from the information contained in the Scopus® database (Elsevier B.V.). These indicators can be used to assess and analyze scientific domains. Citation data are drawn from over 34,100 titles from more than 5000 international publishers and country performance metrics from 239 countries worldwide. Scimago is a Spanish research group from the Consejo Superior de Investigaciones Científicas (CSIC),

Universidad de Granada, Universidad de Extremadura, Universidad Carlos III de Madrid, and Universidad de Alcalá, dedicated to information analysis, representation, and retrieval by means of visualization techniques.

SJR is widely used in research, finding recent work in the areas of sports medicine [32], obstetrics and gynecology [33], and public health [11], among others.

Four public datasets hosted in SJR were selected under the following parameters:

➢ Subject Area: Dentistry.
➢ Subject Category: All subject categories (Dental Assisting, Dental Hygiene, Dental (Miscellaneous), Oral Surgery, Orthodontics, and Periodontics).
➢ Period: 1996–2020.

### 2.2. Statistical Analysis

The indicators used were:

➢ The total number of documents published between 1996 and 2020.
➢ H-index, considering H as the number of documents of a region obtaining at least H citations.
➢ The SJR.
➢ Citations per document in the last 2 years.
➢ Total citations in the last 3 years.

The following region groups were considered:

➢ LAC: This group included the journals indexed in Scimago Journal and Country Rank with a country of origin in Latin America and the Caribbean.
➢ Other regions: This group included the journals indexed in Scimago Journal and Country Rank that are not part of the LAC group.

The normality of the distributions per group of H-Index, SJR, total citations in the past 3 years, and citations per document in the last 2 years, were assessed using the Anderson–Darling normality test. The Flinger–Killen test was used to test the homogeneity of variances. The Wilcoxon rank-sum test with continuity correction was used to compare the mean values between the LAC group and the Other regions group. Finally, the Cullen and Frey graph was used to compare the distributions of the metrics in the skewness–kurtosis space in order to assess the properties of the distribution and to reject unlikely distribution candidates.

### 2.3. Co-Occurrence Network Analysis

A dentistry Co-occurrence Network of the World for the most frequently used keywords was used, which represents the worldwide conceptual structure of research in dentistry through a map. Co-occurrence Networks are useful for displaying and analyzing the intellectual, conceptual, and social structures of research, as well as their evolution and dynamic aspects. The Co-occurrence Network was generated with Bibliometrix [30], which is an open-source software for automating the stages of data analysis and data visualization.

## 3. Results

### 3.1. Publications and Citations

Table 1 shows the countries (30) included in the SJR-All regions/countries (Latin America) category ordered by volume of documents published (1996–2020). In the top five we find Brazil in first place with a total of 29,653 documents (82.94%), Chile in second place with 1389 (3.88%); Mexico in third place with 1081 (3.02%), Colombia in fourth place with 841 documents, and, (2.35%), finally, Cuba, in fifth place with 668 (1.87%). There is a regular and homogeneous volume of production among the countries in second to fifth place (Chile, Mexico, Colombia, and Cuba); however, Brazil (first place) stands out with 82.94% of the documents of the total scientific production compared to 11.12% of the remaining countries in the top five (Chile, Mexico, Colombia, and Cuba).

**Table 1.** Latin American countries with the highest number of publications (1996-2020).

| Rank | Country | Documents | Percentage |
|------|---------|-----------|------------|
| 1 | Brazil | 29,653 | 82.94% |
| 2 | Chile | 1389 | 3.88% |
| 3 | Mexico | 1081 | 3.02% |
| 4 | Colombia | 841 | 2.35% |
| 5 | Cuba | 668 | 1.87% |
| 6 | Argentina | 584 | 1.63% |
| 7 | Peru | 464 | 1.30% |
| 8 | Venezuela | 203 | 0.57% |
| 9 | Puerto Rico | 149 | 0.42% |
| 10 | Uruguay | 116 | 0.32% |
| 11 | Guatemala | 106 | 0.30% |
| 12 | Paraguay | 84 | 0.23% |
| 13 | Ecuador | 83 | 0.23% |
| 14 | Costa Rica | 82 | 0.23% |
| 15 | Trinidad and Tobago | 74 | 0.21% |
| 16 | Jamaica | 54 | 0.15% |
| 17 | Dominican Republic | 43 | 0.12% |
| 18 | El Salvador | 14 | 0.04% |
| 19 | Grenada | 12 | 0.03% |
| 20 | Bolivia | 11 | 0.03% |
| 21 | Panama | 10 | 0.03% |
| 22 | Belize | 9 | 0.03% |
| 23 | Nicaragua | 5 | 0.01% |
| 24 | Haïti | 5 | 0.01% |
| 25 | Netherlands Antilles | 5 | 0.01% |
| 26 | Antigua and Barbuda | 3 | 0.01% |
| 27 | Guyana | 2 | 0.01% |
| 28 | Montserrat | 1 | 0.00% |
| 29 | Virgin Islands (U.S.) | 1 | 0.00% |
| 30 | Barbados | 1 | 0.00% |

Table 2 shows the countries (30) included in the SJR-All regions/countries (Latin America) category ordered by volume of citations (1996–2020). In the top five we find Brazil in first place with a total of 427,968 citations (85.20%), Chile in second place with 18,503 (3.68%), Mexico in third place with 15,044 (3.00%), Colombia in fourth place with 12,055 citations (2.40%), and, finally, Argentina, in fifth place with 9044 citations (1.80%). There is a regular and homogeneous volume of citations between the countries in second to fourth place (Chile, Mexico, and Colombia); however, Brazil (first place) stands out with 85.20% of the citations of the total scientific production.

**Table 2.** Latin American countries with the highest number of citations (1996–2020).

| Rank | Country | Citations | Percentage |
|------|---------|-----------|------------|
| 1 | Brazil | 427,968 | 85.20% |
| 2 | Chile | 18,503 | 3.68% |
| 3 | Mexico | 15,044 | 3.00% |
| 4 | Colombia | 12,055 | 2.40% |
| 5 | Argentina | 9044 | 1.80% |
| 6 | Puerto Rico | 3566 | 0.71% |
| 7 | Peru | 2903 | 0.58% |
| 8 | Venezuela | 2555 | 0.51% |

**Table 2.** *Cont.*

| Rank | Country | Citations | Percentage |
|---|---|---|---|
| 9 | Cuba | 2348 | 0.47% |
| 10 | Guatemala | 2072 | 0.41% |
| 11 | Uruguay | 1412 | 0.28% |
| 12 | Paraguay | 1023 | 0.20% |
| 13 | Trinidad and Tobago | 815 | 0.16% |
| 14 | Costa Rica | 801 | 0.16% |
| 15 | Jamaica | 429 | 0.09% |
| 16 | Ecuador | 270 | 0.05% |
| 17 | Grenada | 249 | 0.05% |
| 18 | Panama | 246 | 0.05% |
| 19 | Dominican Republic | 187 | 0.04% |
| 20 | El Salvador | 171 | 0.03% |
| 21 | Belize | 162 | 0.03% |
| 22 | Netherlands Antilles | 134 | 0.03% |
| 23 | Bolivia | 120 | 0.02% |
| 24 | Haïti | 70 | 0.01% |
| 25 | Guyana | 47 | 0.01% |
| 26 | Antigua and Barbuda | 32 | 0.01% |
| 27 | Virgin Islands (U.S.) | 25 | 0.00% |
| 28 | Montserrat | 21 | 0.00% |
| 29 | Nicaragua | 19 | 0.00% |
| 30 | Barbados | 0 | 0.00% |

*3.2. Journals*

There are 201 dentistry journals indexed in Scimago Journal and Country Rank in 2020, of which there are 12 journals from the LAC region. Of these 12 journals, only *Pesquisa odontologica brasileira = Brazilian oral research* (Brazil) and *Journal of Applied Oral Science* (Brazil) are in the first quartile (Q1). Table 3 shows the dentistry journals from LAC listed in Scimago Journal and Country Rank (1996–2020). Furthermore, 10 of the 12 LAC journals belong to Brazilian universities or research institutes, and the remaining two to Chilean (1) and Cuban (1) institutions.

*3.3. H-Index Comparison*

The Anderson–Darling normality test proved that the H-Index distributions corresponding to the LAC region and the Other regions group were not normally distributed, with $p < 0.01$ in both cases. The Cullen and Frey graph (Figure 1) shows that the distribution of the H-Index of journals from the LAC region cannot be explained by the normal, negative binomial, or Poisson distributions, with min = 1, max = 50, median = 9, mean = 17.83, sd = 18.01, skewness = 1.07, and kurtosis = 2.38. On the other hand, this graph suggests that the H-Index of journals from other regions is close to a Poisson distribution, with min = 1, max = 182, median = 31, mean = 42.01, sd = 39.28, skewness = 1.25, and kurtosis = 4.08.

The Fligner–Killeen test of homogeneity of variances proved that the variances of the H-Index of LAC and other regions are not equal ($p = 0.0256$); therefore, we can only test for differences in the mean of the two groups. The Wilcoxon rank-sum test indicated that the H-Index of other regions is statistically significantly higher than the H-Index of LAC, with $W = 702$, $p = 0.0246$. The difference in the distribution of the H-Index between LAC and other regions is displayed in Figure 2. Figure 3 shows the global comparison of the H-Index.

**Table 3.** Latin American Journals in the Dentistry Area (1996–2020).

| Rank | Title | SJR | SJR Best Quartile | H Index | Total Docs. (3 Years) | Total Citations (3 Years) | Country | Publisher |
|---|---|---|---|---|---|---|---|---|
| 1 | *Pesquisa odontologica brasileira = Brazilian oral research* | 0.847 | Q1 | 45 | 361 | 961 | Brazil | University of Sao Paolo |
| 2 | *Journal of Applied Oral Science* | 0.754 | Q1 | 44 | 281 | 703 | Brazil | Faculdade de Odontologia de Bauru |
| 3 | *Brazilian Dental Journal* | 0.616 | Q2 | 50 | 282 | 498 | Brazil | Associacao Brasileira de Divulgacao Cientifica |
| 4 | *Dental Press Journal of Orthodontics* | 0.57 | Q2 | 22 | 210 | 345 | Brazil | Dental Press Editora Ltd.a |
| 5 | *Pesquisa Brasileira em Odontopediatria e Clinica Integrada* | 0.185 | Q4 | 12 | 340 | 450 | Brazil | Association of Support to Oral Health Research (APESB) |
| 6 | *Brazilian Dental Science* | 0.153 | Q4 | 6 | 221 | 216 | Brazil | Universidade Estadual Paulista |
| 7 | *Journal of Oral Research* | 0.127 | Q4 | 4 | 242 | 53 | Chile | Universidad de Concepcion |
| 8 | *Brazilian Journal of Oral Sciences* | 0.125 | Q4 | 11 | 144 | 28 | Brazil | Universidade Estadual de Campinas |
| 9 | *Revista Cubana de Estomatologia* | 0.124 | Q4 | 7 | 151 | 24 | Cuba | Editorial Ciencias Medicas |
| 10 | *Revista Odonto Ciencia* | 0.111 | Q4 | 7 | 64 | 14 | Brazil | Pontificia Universidade Catolica do Rio Grande do Sul |
| 11 | *Revista Clinica de Ortodontia Dental Press* | 0.106 | Q4 | 1 | 81 | 2 | Brazil | Dental Press International |
| 12 | *Dental Press Endodontics* | 0.1 | Q4 | 5 | 72 | 1 | Brazil | Dental Press Editora Ltd.a |

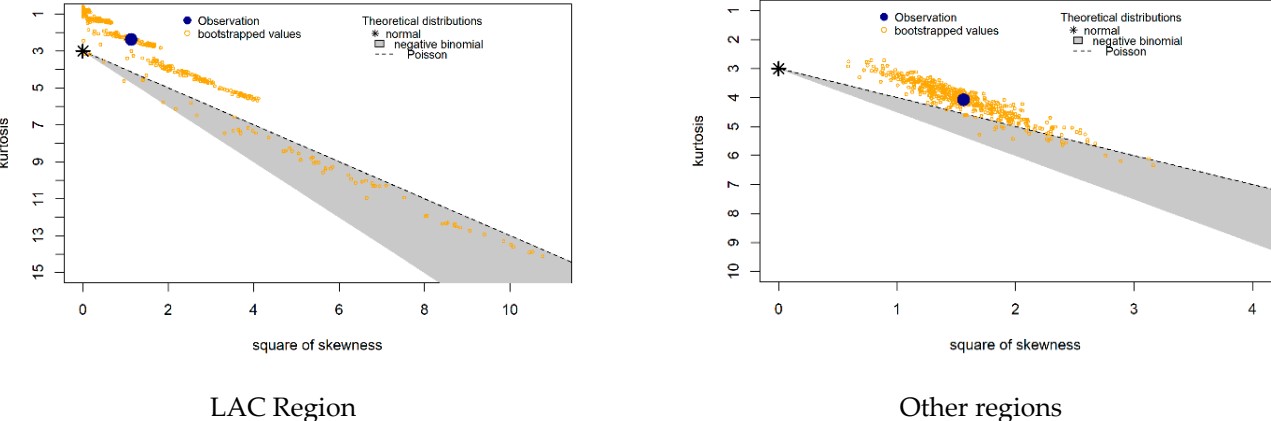

LAC Region             Other regions

**Figure 1.** Cullen and Frey graph of the H-Index of journals from LAC and other regions.

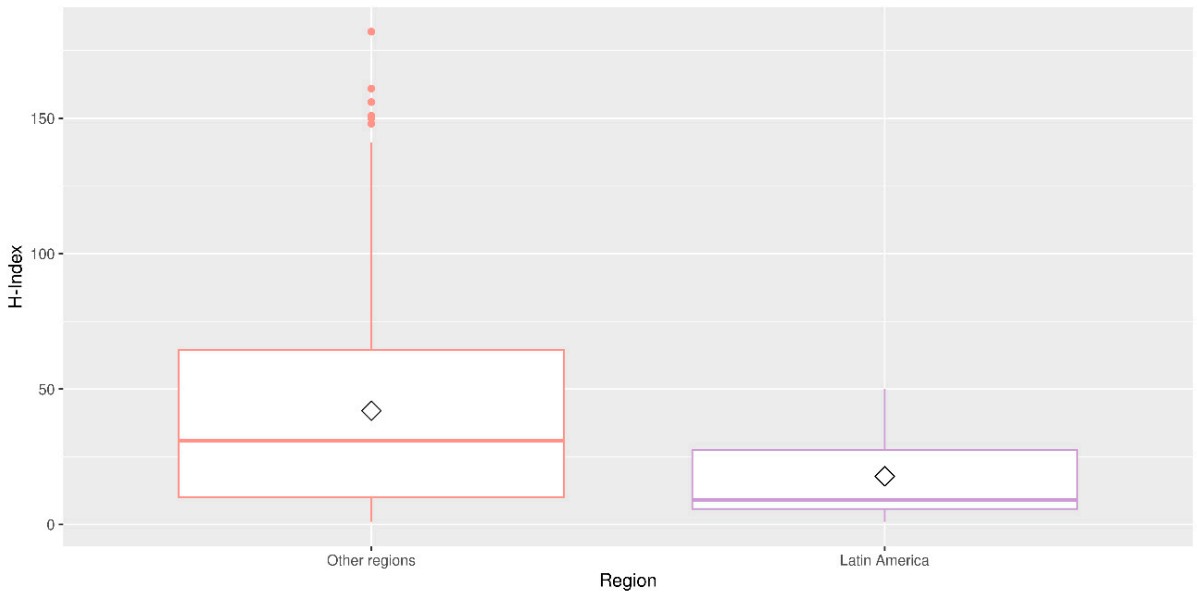

**Figure 2.** Comparison of H-Index between LAC and other regions.

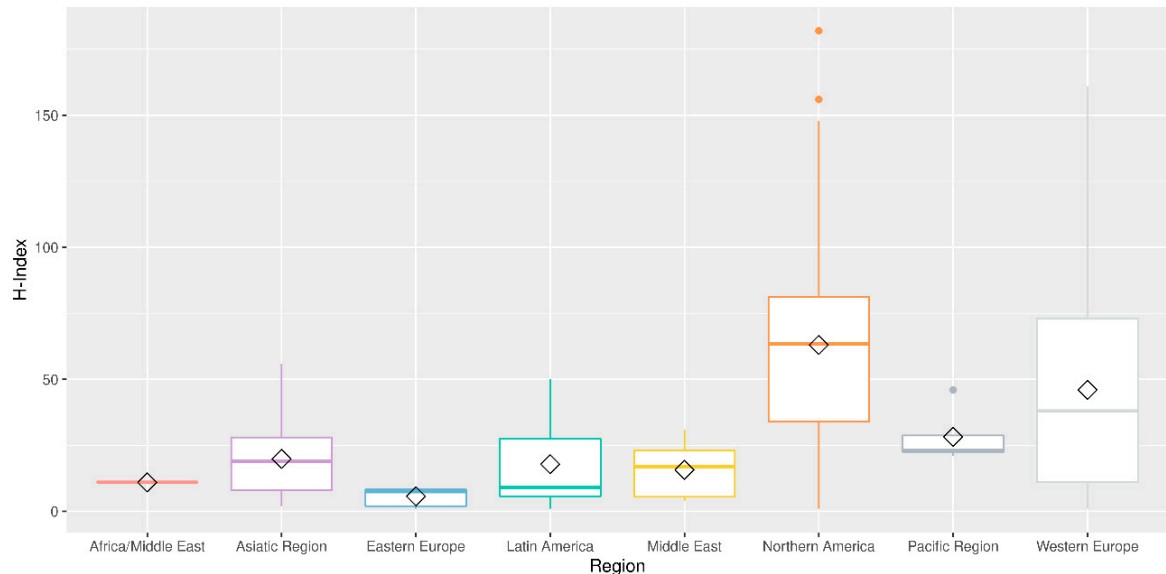

**Figure 3.** Global comparison of H-Index between all regions.

### 3.4. SJR Comparison

The Anderson–Darling normality test proves that the SJR distributions corresponding to the LAC and other regions groups were not normally distributed, with $p < 0.001$ in both cases. The Cullen and Frey graph (Figure 4) shows that the distribution of the SJR of Journals of the LAC region can be explained by a beta distribution, with min = 0.1, max = 0.85, median = 0.14, mean = 0.32, sd = 0.29, skewness = 0.98, and kurtosis = 2.11. On the other hand, this graph suggests that the SJR of journals from other regions is close to a lognormal distribution, with min = 0.1, max = 3.73, median = 0.47, mean = 0.60, sd = 0.54, skewness = 2.56, and kurtosis = 12.98.

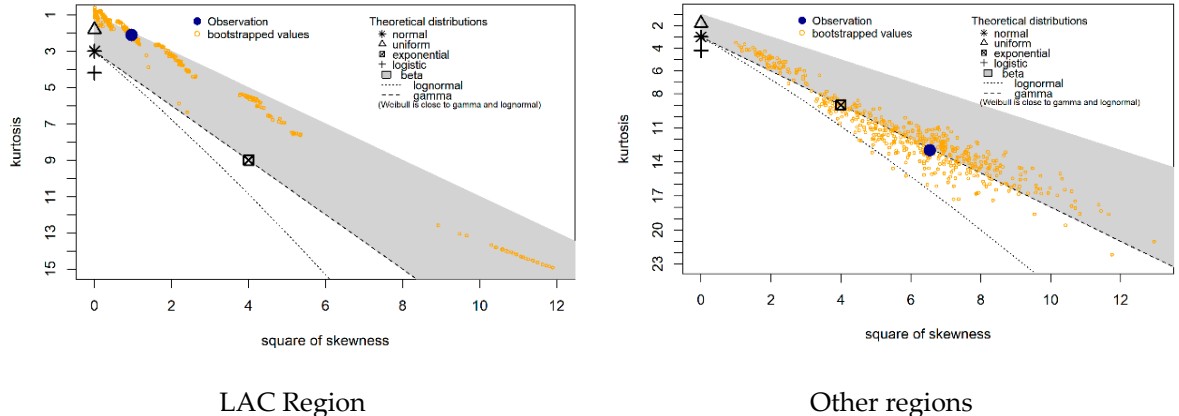

LAC Region　　　　　　　　　　　　　　　　　　　　　　Other regions

**Figure 4.** Cullen and Frey graph of the SJR of journals from LAC and other regions.

The Fligner–Killeen test of homogeneity of variances proved that the variances of SJR of LAC and other regions are homogeneous, with $p = 0.1168$. The Wilcoxon rank-sum test indicated that the SJR of LAC is statistically significantly lower than the SJR of other regions, with $W = 651$, $p = 0.0135$. Figure 5 shows a graphical representation of the SJR Comparison between both groups. Figure 6 shows the global comparison of SJR.

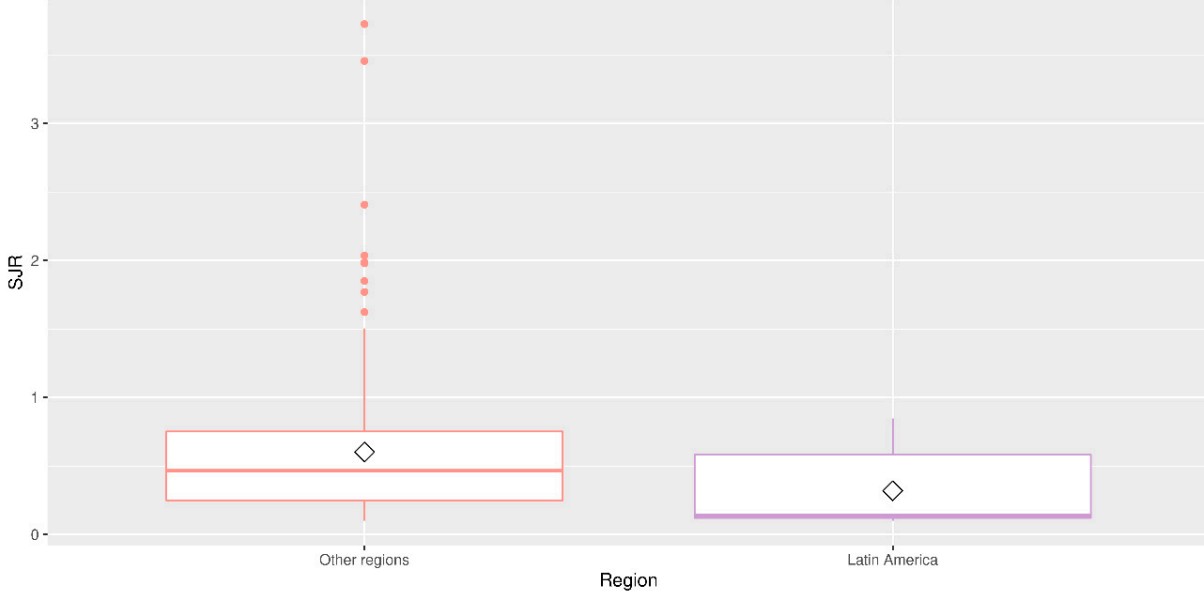

**Figure 5.** Comparison of SJR between LAC and other regions.

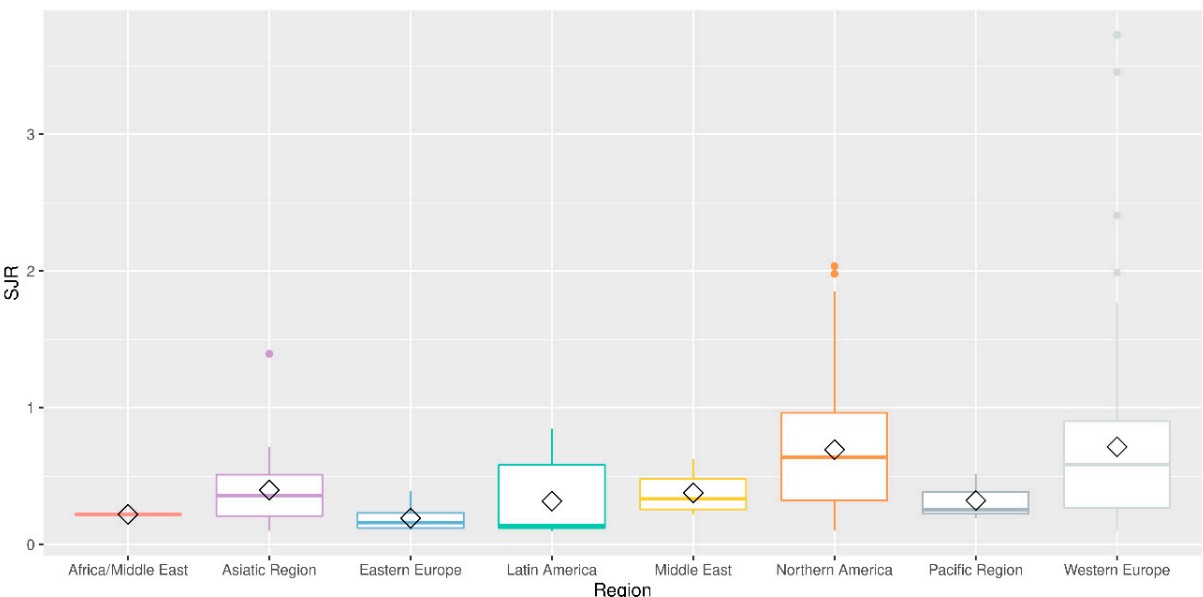

**Figure 6.** Global comparison of SJR between all regions.

*3.5. Citations per Document in the Last 2 Years Comparison*

The Anderson–Darling normality test proved that the citations per document distributions corresponding to the LAC and other regions groups were not normally distributed, with $p < 0.04$ in both cases. The Cullen and Frey graph (Figure 7) shows that the distribution of the citations per document of journals of the LAC can be explained by a beta distribution, with min = 0.02, max = 2.36, median = 1.67, mean = 0.93, sd = 0.90, skewness = 0.48, and kurtosis = 1.55. On the other hand, this graph suggests that the citations per document of journals from other regions is close to a lognormal distribution, with min = 0, max = 7.37, median = 1.41, mean = 1.70, sd = 1.41, skewness = 1.42, and kurtosis = 5.56.

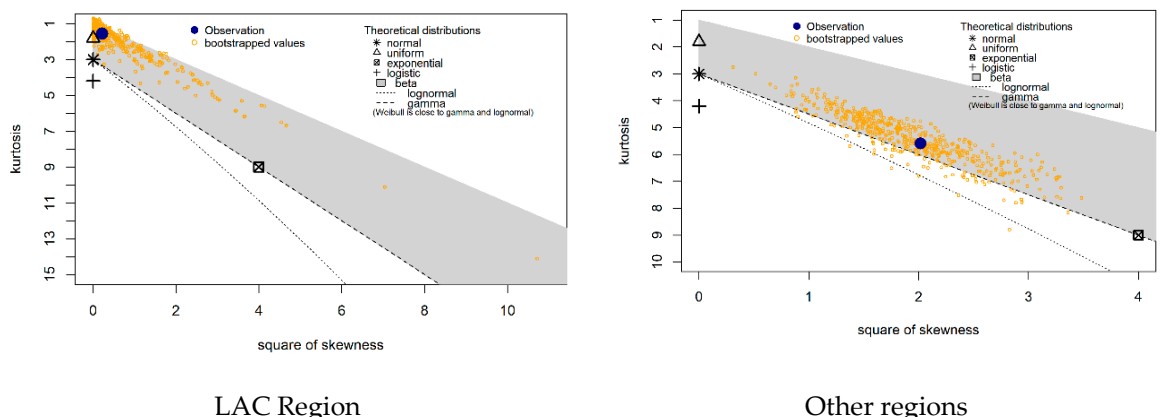

LAC Region                                    Other regions

**Figure 7.** Cullen and Frey graph of the citations per document of journals from LAC and other regions.

The Fligner–Killeen test of homogeneity of variances proved that the variances of citations per document of LAC and other regions are homogeneous, with $p = 0.4751$. The Wilcoxon rank-sum test proved that there are no differences in the citations per document between LAC and other regions, with $W = 770$, $p = 0.057$. Figure 8 shows a graphical representation of the comparison of citations per document between both groups. Figure 9 shows the global comparison of citations per document.

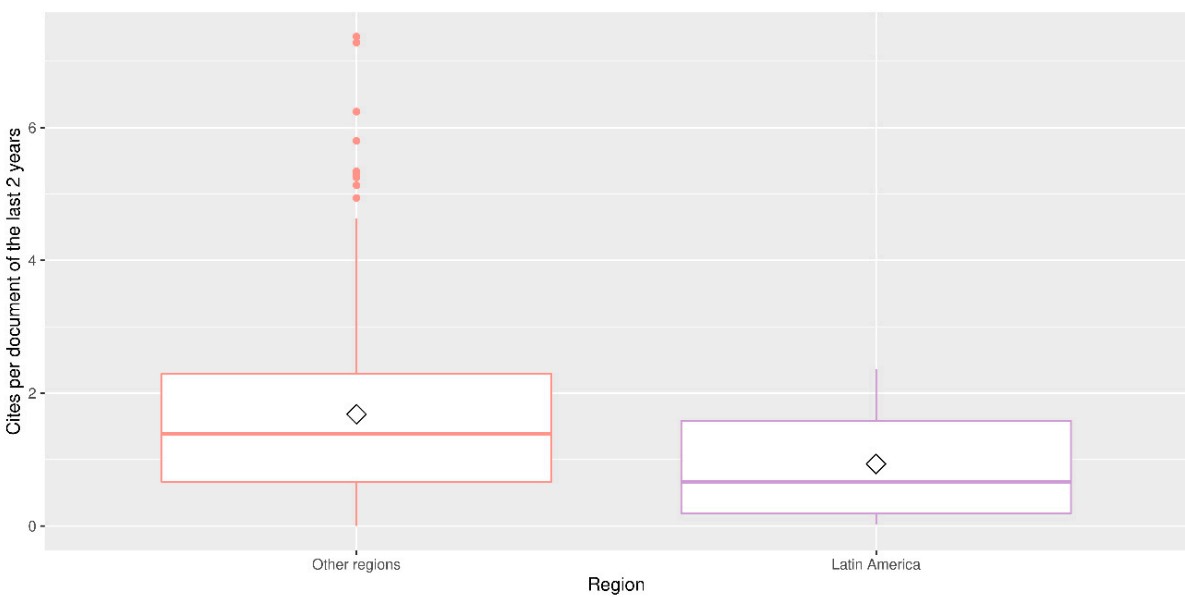

**Figure 8.** Comparison of citations per document between LAC and other regions.

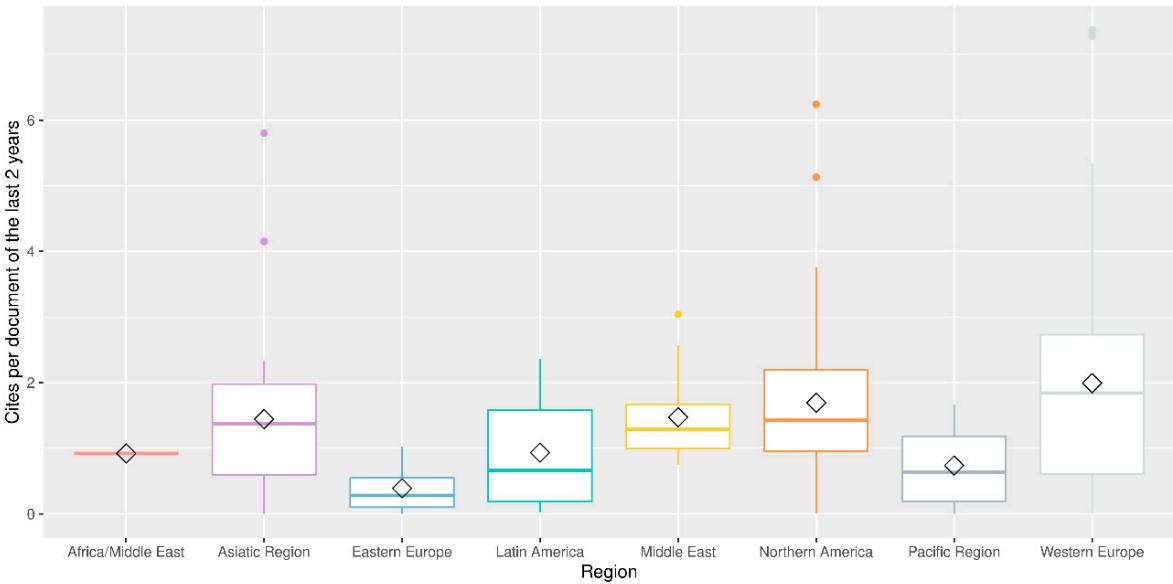

**Figure 9.** Global comparison of citations per document between all regions.

### 3.6. Comparison of the Total Citations in the Last 3 Years

The Anderson–Darling normality test proved that the distributions of total citations corresponding to the LAC and other regions groups were not normally distributed, with $p < 0.03$ in both cases. The Cullen and Frey graph (Figure 10) shows that the distribution of the total citations in the last 3 years of journals from the LAC region cannot be explained by the normal, negative binomial, or Poisson distributions, with min = 1, max = 961, median = 135, mean = 274.58, sd = 231.01, skewness = 1.05, and kurtosis = 3.19. On the other hand, this graph suggests that the total citations in the last 3 years of journals from other regions is close to a Poisson distribution, with min = 0, max = 3682, median = 286, mean = 569.14, sd = 788.59, skewness = 2.36, and kurtosis = 8.50.

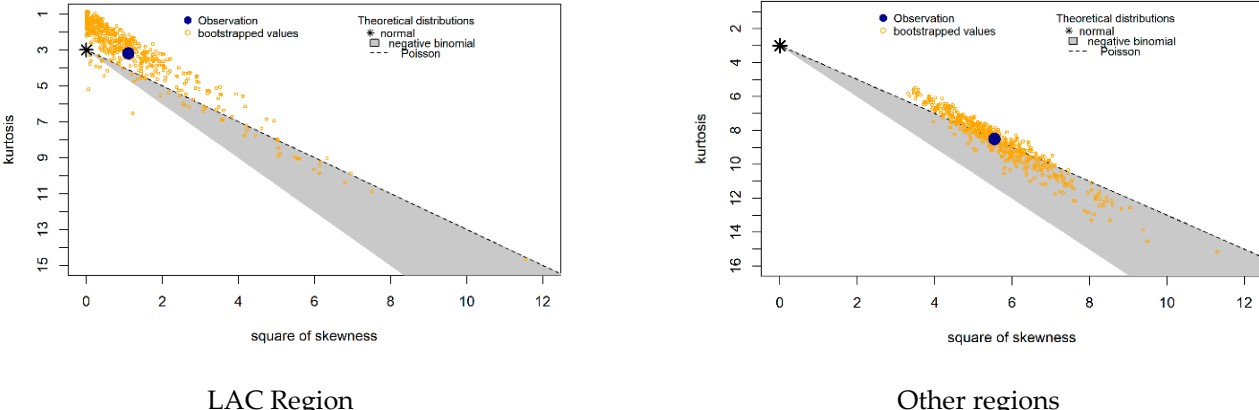

LAC Region                                    Other regions

**Figure 10.** Cullen and Frey graph of total citations in the last 3 years for LAC journals and other regions.

The Fligner–Killeen test of homogeneity of variances proved that the variance of total citations of LAC and other regions is homogeneous, with $p$ = 0.3028. The Wilcoxon rank-sum test proved that there are no differences in the total citations between LAC and other regions, with $W$ = 866, and $p$ = 0.1568. Figure 11 shows a graphical representation of the total citations comparison between both groups. Figure 12 shows a global comparison of total citations.

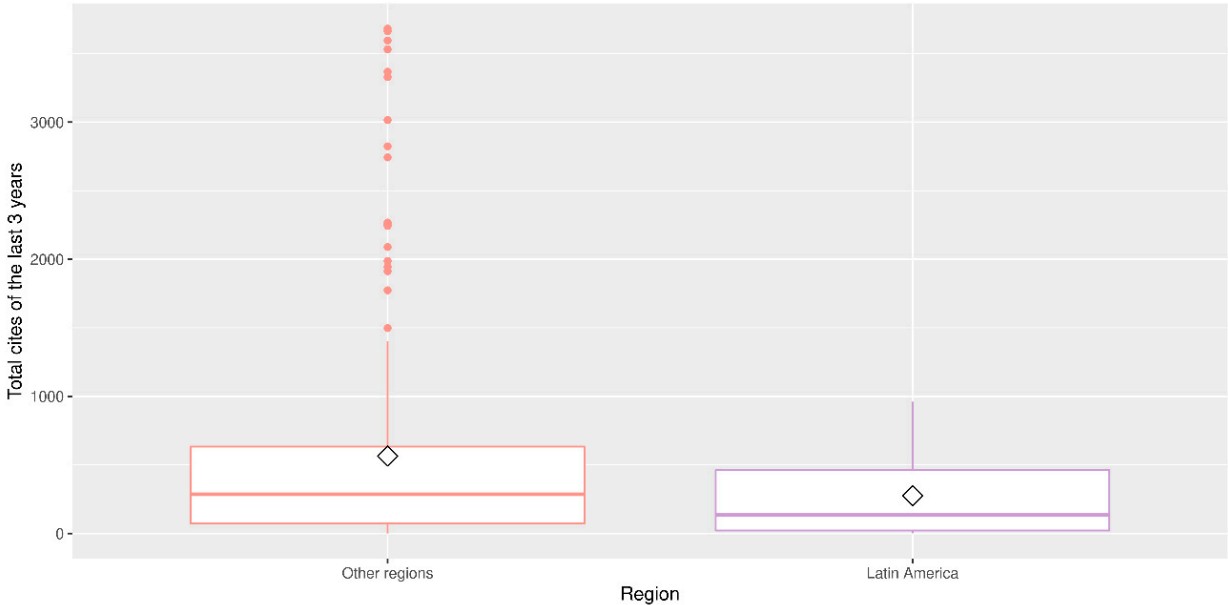

**Figure 11.** Comparison of total citations between LAC and other regions.

*3.7. Relationship between Journals and Institutions in LAC*

The rankings of the publishers, as academic institutions, provide insights into the quality of the integration of academic activities and scientific research. Table 4 provides a summary of the relationships between the dentistry journals and research institutions in LAC, alongside the Academic Ranking of Global Universities (ARWU) 2020 and QS 2020 metrics. The results show that Brazil's top three dentistry journals in LAC are related to Brazilian universities ranked in the top 101–150 positions of the ARWU 2020 [34] and the top 116 of the QS 2020 (QS World University Rankings 2020: Top Global Universities | Top Universities, n.d.) [35] indices.

**Table 4.** Latin American Journals in the Dentistry Area and ARWU 2020 and QS 2020 Ranking (1996–2020).

| Rank | Title | Country | Publisher | ARWU 2020 | QS 2020 |
|---|---|---|---|---|---|
| 1 | *Pesquisa odontologica brasileira = Brazilian oral research* | Brazil | University of Sao Paolo | 101–150 | 116 |
| 2 | *Journal of Applied Oral Science* | Brazil | Faculdade de Odontologia de Bauru (University of Sao Paolo) | 101–150 | 116 |
| 3 | *Brazilian Dental Journal* | Brazil | Fundação Odontológica de Ribeirão Preto da Faculdade de Odontologia de Ribeirão Preto da Universidade de São Paulo | 101–150 | 116 |
| 4 | *Dental Press Journal of Orthodontics* | Brazil | Dental Press Editora Ltd.a | | |
| 5 | *Pesquisa Brasileira em Odontopediatria e Clinica Integrada* | Brazil | Association of Support to Oral Health Research (APESB) | | |
| 6 | *Brazilian Dental Science* | Brazil | Universidade Estadual Paulista, Institute of Science and Technology of Sao Jose dos Campo | 301–400 | 482 |
| 7 | *Journal of Oral Research* | Chile | Universidad de Concepcion | 801–900 | 601–650 |
| 8 | *Brazilian Journal of Oral Sciences* | Brazil | Universidade Estadual de Campinas | 301–400 | 214 |
| 9 | *Revista Cubana de Estomatologia* | Cuba | Editorial Ciencias Medicas | | |
| 10 | *Revista Odonto Ciencia* | Brazil | Pontificia Universidade Catolica do Rio Grande do Sul | | 801–1000 |
| 11 | *Revista Clinica de Ortodontia Dental Press* | Brazil | Dental Press International | | |
| 12 | *Dental Press Endodontics* | Brazil | Dental Press Editora Ltd.a | | |

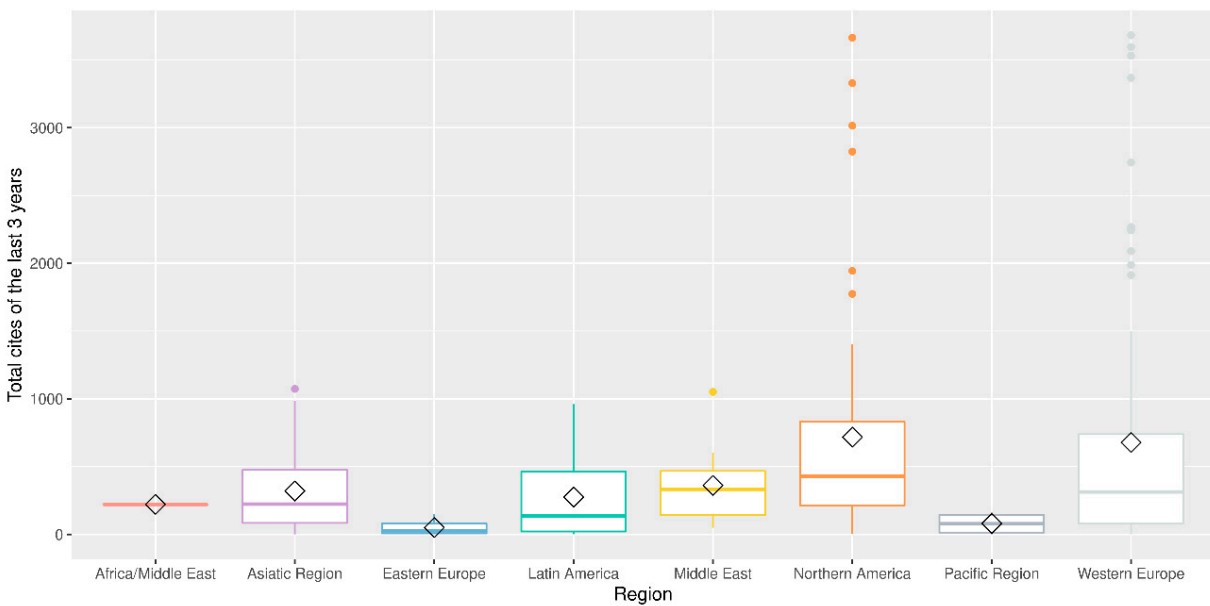

**Figure 12.** Global comparison of total citations between all regions.

Figure A1 in Appendix A shows the World Dentistry Co-occurrence Network generated with Bibliometrix [30], which represents the worldwide conceptual structure of research in dentistry through a co-occurrence map of keywords during the period 1996-to 2020. Three clusters can be appreciated: the largest one links the keywords "human", "humans", "methodology", and "review"; the second one links the keywords "article", "male", "female", "adult", "middle-aged", and "periodontal disease"; finally, the third cluster links the keywords "tooth implantation", "dental implants", and "tooth prosthesis". Similarly, Figure A2 shows the Latin American Dentistry Co-occurrence Network (1996–2020) generated with Bibliometrix, where three clusters can be appreciated: the largest one links the keywords "human", "humans", "article", "female", "male", and "controlled study". The second cluster links the keywords "comparative study", "chemistry", "methodology", "dentin", and "material testing. Finally, the smallest cluster links the keywords "statistics", "animals", "drug effect", "chlorhexidine", and "endodontics".

Furthermore, the World Dentistry Thematic Map and the Latin America Dentistry Thematic Map are shown in Figures A3 and A4, respectively, in Appendix A. In both cases, the major themes include "human", "article", "review", "male", and "female". The emerging or declining themes for the world include "computer-assisted models", "tomography", "tooth radiology", and "imaging". On the other hand, the emerging or declining themes in LAC are "animals", "physiology", "dental implants", and "alveolar bone loss".

## 4. Discussion

### 4.1. Publications and Citations

According to data from Scimago, Brazil has the largest research output in LAC; furthermore, it is the country in LAC with the largest number of dental research articles published yearly and the second country in the world in the number of published dental articles since 2006 [36]. Brazil is currently considered the dental powerhouse of LAC, with a growing dental industry. According to the economic report published by the Brazilian Association of the Medical Devices Industry (ABIMO), the main dental exports of Brazil are bio-ceramic cement, orthodontic appliances, and dental implants; furthermore, the overall dental industry exports of Brazil increased by 38.96% in 2020 [37]. Brazil is one of the most promising emerging economies [38] and a member of the BRICS countries [39,40].

Brazil has one of the highest GDPs (Intl\$ 3,328,459 million) and GDPs per capita (Intl\$ 15,642) of LAC according to the gross domestic product (nominal) provided by the

International Monetary Fund's estimates in the October 2020 World Economic Outlook database. Furthermore, these indicators can be related to the quality (citations), volume (publications), and dissemination of scientific production [41,42].

The Regional Cooperative Online Information System for Scholarly Journals from Latin America, the Caribbean, Spain and Portugal (LATINDEX) is an initiative to promote the scientific integration of the region, consisting of a large bibliographical information system available for free for consultation, with a directory of over 450 scholarly dentistry journals in Spanish and Portuguese. The results of this study suggest that there is a vast difference in the number of dentistry journals published in Spanish and Portuguese listed in LATINDEX compared to those indexed in the Scimago Journal & Country Rank.

### 4.2. Journals

The results show that Brazil has the hegemony in scientific publications related to dentistry in the Latin American ecosystem (Table 4). Twelve journals are indexed in SJR and ten belong to Brazil. This indicates that, despite occupying the hegemony of scientific publications in Latin America, Brazil has few well-positioned journals. Likewise, the University of Sao Paulo is the one with the most extensive scientific production. The institution also has associated research centers (foundations and associations) that produce a large number of scientific papers.

It is noteworthy that most of the indexed journals are published by independent publishers not linked to any university or research center. In addition, the indexed journals belong to institutions of higher education that occupy relevant positions at the international level. More than 80% of the institutions mentioned occupy relevant positions (1–500) in the different rankings of prestige and quality of higher education such as ARWU [34] and QS [35].

### 4.3. Metrics Analysis

Regarding the H-Index comparison between the LAC region and the rest of the world, the results show the journals located in LAC countries have a significantly lower mean H-Index compared to other regions. These findings suggest that the impact of dental research emerging from Latin America and the Caribbean is low from a worldwide perspective. Although this study cannot fully explain the reasons behind these findings, it is important to notice that the language barrier might be an important obstacle for academics and researchers to publish their discoveries in the English language.

In the case of the number of citations, the distributions of citations for both the LAC region and the rest of the world are not normally distributed. This matches previous research that suggests that the distribution of bibliometric indicators rarely follows a normal distribution but can be well represented by other models such as the lognormal function [43]. This phenomenon indicates that scholars that publish more have a higher probability of being recognized, which can help to explain the large difference in published documents and the number of citations between Brazil and the rest of the LAC region.

Academic institutions, governments, and funding providers heavily rely on metrics to support decisions that impact research opportunities; however, this has led to a massive misapplication of indicators in the evaluation of scientific performance. In this regard, the third principle of the Leiden Manifesto for research metrics encourages the protection of excellence in locally relevant research [44]. In this regard, the Co-occurrence Network of keywords for both the world and Latin America resulted in three clusters, and the main topics are shared between both networks. Therefore, these results suggest that the topics covered in dental research articles from LAC are comparable to the rest of the world. However, in the case of LAC, topics that are locally relevant in dental research may have not reached widespread attention from the scientific community.

Furthermore, high-impact Spanish-language dentistry papers may not have been considered in this research because only a very small number of dentistry journals from

LAC that publish papers in the English language are indexed in Scopus and listed in the SCImago journal and country rank.

## 5. Conclusions

This research found evidence that the impact of the production of scientific journals specializing in dental research in Latin America and the Caribbean is lower than in the rest of the world, with lower H-Index and SJR metrics, thus disconfirming the $H1_0$ and $H2_0$ null hypotheses of the study. In contrast, there were no differences in the mean number of citations per document and the mean total number of citations between LAC and other regions; therefore, the findings of this research could not allow us to reject $H3_0$ and $H4_0$. In this regard, even if the articles published in dentistry journals from LAC are being cited in similar proportions to the journals of other regions, a large portion of these citations originated from publications with low scientific impact. Furthermore, the conceptual structure of keywords related to dental research in LAC is comparable to the global trends; therefore, the causes behind this difference in impact cannot be linked to discrepancies in the research topics. On the other hand, previous research has suggested that some of the underlying causes of the differences in journal metrics between LAC and the rest of the world might be related to the ranking of the institutions or universities that govern the journals, the low availability of resources for scientific research, the low perceived importance of dentistry, and the language barrier [45–48].

### *Future Research Lines*

In this research, the analysis of publications by the author has not been considered, nor have they been analyzed by specialty or subject matter; it would be interesting for future studies to focus on this topic of research. Likewise, it would be interesting to study other regions of the world individually in this same category in order to obtain the current radiography of research in dentistry around the world, such as a direct comparison of the research output between the LAC and North America regions [49], and to deepen patterns of scientific collaboration in dentistry between regions [50,51].

This study was centered around journals indexed by Scopus, where most of the journals of the region originated from Brazil; therefore, future in-depth analysis of the scientific production in dental research in the LAC region should consider journals indexed in regional bibliographical information systems such as LATINDEX. Furthermore, to better address the analysis of scientific production in Latin America and the Caribbean, each country should be studied separately and normalize its publication metrics by population and socioeconomic metrics.

**Author Contributions:** Conceptualization, G.V. and P.S.-N.; methodology, G.V.; software, P.S.-N.; validation, P.S.-N., P.W.-R. and G.V.; formal analysis, G.V.; investigation, P.S.-N.; resources, P.W.-R.; data curation, G.V.; writing—original draft preparation, P.S.-N.; writing—review and editing, P.S.-N. and P.W.-R.; visualization, G.V. and P.S.-N.; supervision, P.W.-R.; project administration, P.S.-N. All authors have read and agreed to the published version of the manuscript.

**Funding:** This research received no external funding.

**Data Availability Statement:** All the data used for this research are publicly available and provided by Scimago at https://www.scimagojr.com accessed on 19 December 2021.

**Conflicts of Interest:** The authors declare no conflict of interest.

## Appendix A

Figure A1 represents the worldwide conceptual structure of research in dentistry through a co-occurrence map of keywords during the period 1996–2020.

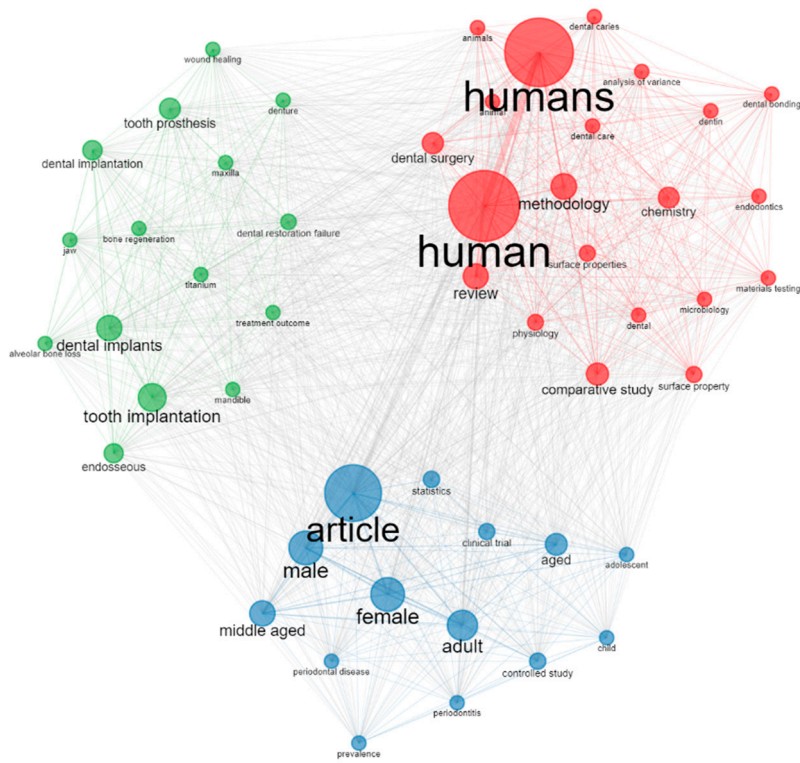

**Figure A1.** Conceptual Structure. World Dentistry Co-occurrence Network (1996–2020) generated with Bibliometrix [30]. Sort on: Cited by (highest) first 2000 document results retrieved. Field: Keyword Plus, Network Parameters: Network Layout (Automatic layout), Normalization (Association), Number of nodes (50), Clustering Algorithm (Louvain), Node Color by Year (No), Repulsion Force (0,1), Minimum number of edges (2), Remove isolated nodes (Yes). Graphical parameters: Opacity (0,7), Label cex (Yes), Label Size (Yes), Node Shadow (No), Number of labels (50), Node Shape (Dot), Edge Size (5), and Curved edges (No). Scopus Search Query: SUBJAREA (dent) AND (LIMIT-TO (PUBYEAR, 2020) OR LIMIT-TO (PUBYEAR, 2019) OR LIMIT-TO (PUBYEAR, 2018) OR LIMIT-TO (PUBYEAR, 2017) OR LIMIT-TO (PUBYEAR, 2016) OR LIMIT-TO (PUBYEAR, 2015) OR LIMIT-TO (PUBYEAR, 2014) OR LIMIT-TO (PUBYEAR, 2013) OR LIMIT-TO (PUBYEAR, 2012) OR LIMIT-TO (PUBYEAR, 2011) OR LIMIT-TO (PUBYEAR, 2010) OR LIMIT-TO (PUBYEAR, 2009) OR LIMIT-TO (PUBYEAR, 2008) OR LIMIT-TO (PUBYEAR, 2007) OR LIMIT-TO (PUBYEAR, 2006) OR LIMIT-TO (PUBYEAR, 2005) OR LIMIT-TO (PUBYEAR, 2004) OR LIMIT-TO (PUBYEAR, 2003) OR LIMIT-TO (PUBYEAR, 2002) OR LIMIT-TO (PUBYEAR, 2001) OR LIMIT-TO (PUBYEAR, 2000) OR LIMIT-TO (PUBYEAR, 1999) OR LIMIT-TO (PUBYEAR, 1998) OR LIMIT-TO (PUBYEAR, 1997) OR LIMIT-TO (PUBYEAR, 1996)). Search date on Scopus database: 06 June 2022. Search: 334,555 document results.

Figure A2 represents the Latin American conceptual structure of research in dentistry through a co-occurrence map of keywords during the period 1996–2020.

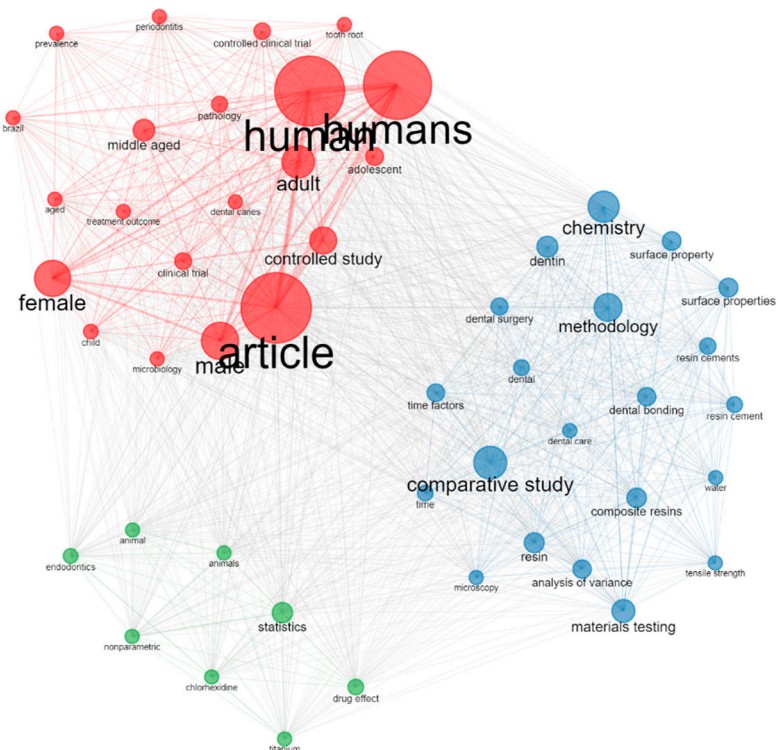

**Figure A2.** Conceptual Structure. Latin American Dentistry Co-occurrence Network (1996–2020) generated with Bibliometrix [30]. Sort on: Cited by (highest) first 2000 document results retrieved. Field: Keyword Plus, Network Parameters: Network Layout (Automatic layout), Normalization (Association), Number of nodes (50), Clustering Algorithm (Louvain), Node Color by Year (No), Repulsion Force (0,1), Minimum number of edges (2), Remove isolated nodes (Yes). Graphical parameters: Opacity (0,7), Label cex (Yes), Label Size (Yes), Node Shadow (No), Number of labels (50), Node Shape (Dot), Edge Size (5), and Curved edges (No). Scopus Search Query: SUBJAREA (dent) AND (LIMIT-TO (AFFILCOUNTRY, "Brazil") OR LIMIT-TO (AFFILCOUNTRY, "Chile") OR LIMIT-TO (AFFILCOUNTRY, "Mexico") OR LIMIT-TO (AFFILCOUNTRY, "Colombia") OR LIMIT-TO (AFFIL-COUNTRY, "Argentina") OR LIMIT-TO (AFFILCOUNTRY, "Cuba") OR LIMIT-TO (AFFILCOUNTRY, "Peru") OR LIMIT-TO (AFFILCOUNTRY, "Venezuela") OR LIMIT-TO (AFFILCOUNTRY, "Puerto Rico") OR LIMIT-TO (AFFILCOUNTRY, "Uruguay") OR LIMIT-TO (AFFILCOUNTRY, "Guatemala") OR LIMIT-TO (AFFILCOUNTRY, "Ecuador") OR LIMIT-TO (AFFILCOUNTRY, "Costa Rica") OR LIMIT-TO (AFFILCOUNTRY, "Paraguay") OR LIMIT-TO (AFFILCOUNTRY, "Trinidad and Tobago") OR LIMIT-TO (AFFILCOUNTRY, "Dominican Republic") OR LIMIT-TO (AFFILCOUNTRY, "Jamaica") OR LIMIT-TO (AFFILCOUNTRY, "Panama") OR LIMIT-TO (AFFILCOUNTRY, "Belize") OR LIMIT-TO (AFFILCOUNTRY, "Nicaragua")) AND (LIMIT-TO (PUBYEAR, 2020) OR LIMIT-TO (PUBYEAR, 2019) OR LIMIT-TO (PUBYEAR, 2018) OR LIMIT-TO (PUBYEAR, 2017) OR LIMIT-TO (PUBYEAR, 2016) OR LIMIT-TO (PUBYEAR, 2015) OR LIMIT-TO (PUBYEAR, 2014) OR LIMIT-TO (PUBYEAR, 2013) OR LIMIT-TO (PUBYEAR, 2012) OR LIMIT-TO (PUBYEAR, 2011) OR LIMIT-TO (PUBYEAR, 2010) OR LIMIT-TO (PUBYEAR, 2009) OR LIMIT-TO (PUBYEAR, 2008) OR LIMIT-TO (PUBYEAR, 2007) OR LIMIT-TO (PUBYEAR, 2006) OR LIMIT-TO (PUBYEAR, 2005) OR LIMIT-TO (PUBYEAR, 2004) OR LIMIT-TO (PUBYEAR, 2003) OR LIMIT-TO (PUBYEAR, 2002) OR LIMIT-TO (PUBYEAR, 2001) OR LIMIT-TO (PUBYEAR, 2000) OR LIMIT-TO (PUBYEAR, 1999) OR LIMIT-TO (PUBYEAR, 1998) OR LIMIT-TO (PUBYEAR, 1997) OR LIMIT-TO (PUBYEAR, 1996)) Search date on Scopus database: 6 June 2022. Search: 35,032 document results.

Figure A3 represents the worldwide conceptual structure of research in dentistry through a thematic map of keywords during the period 1996–2020. The figure shows the quadrants of motor themes, niche themes, emerging or declining themes, and basic themes.

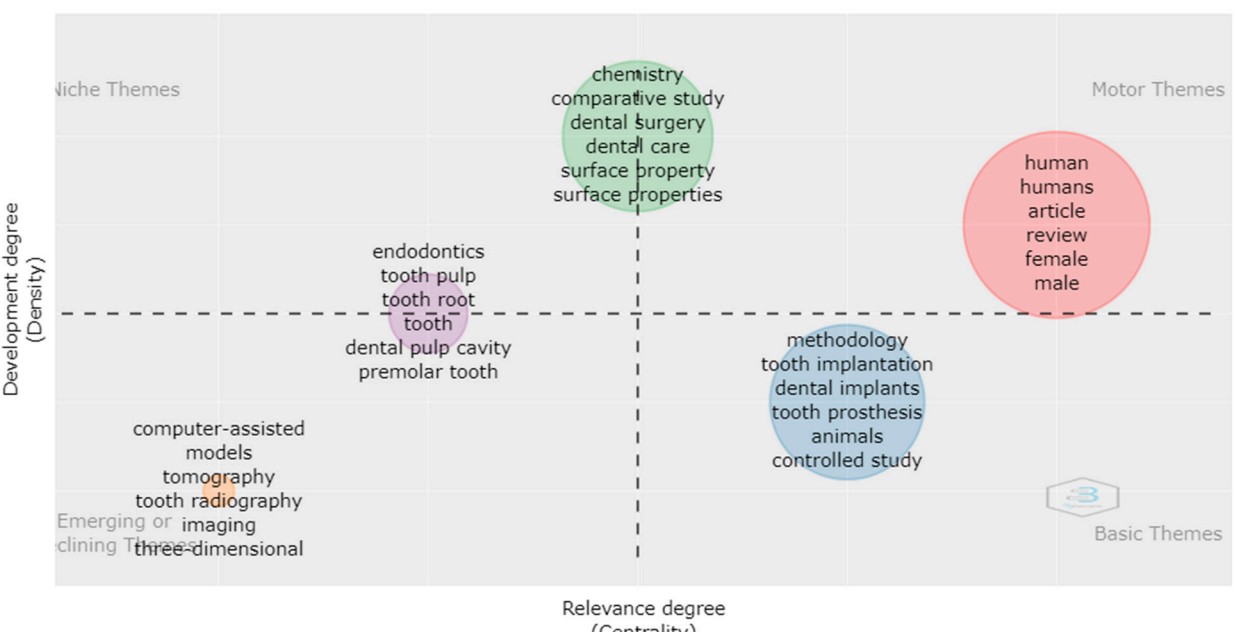

**Figure A3.** Conceptual Structure. World Dentistry Thematic Map (1996–2020) generated with Bibliometrix [30]. Sort on: Cited by (highest)-First 2000 document results retrieved. Field: Keyword Plus, Number of words (560), Min cluster frequency (1–5), Number of labels (0–6), and Label size (0–0,3). Scopus Search Query: SUBJAREA (dent) AND (LIMIT-TO (PUBYEAR, 2020) OR LIMIT-TO (PUBYEAR, 2019) OR LIMIT-TO (PUBYEAR, 2018) OR LIMIT-TO (PUBYEAR, 2017) OR LIMIT-TO (PUBYEAR, 2016) OR LIMIT-TO (PUBYEAR, 2015) OR LIMIT-TO (PUBYEAR, 2014) OR LIMIT-TO (PUBYEAR, 2013) OR LIMIT-TO (PUBYEAR, 2012) OR LIMIT-TO (PUBYEAR, 2011) OR LIMIT-TO (PUBYEAR, 2010) OR LIMIT-TO (PUBYEAR, 2009) OR LIMIT-TO (PUBYEAR, 2008) OR LIMIT-TO (PUBYEAR, 2007) OR LIMIT-TO (PUBYEAR, 2006) OR LIMIT-TO (PUBYEAR, 2005) OR LIMIT-TO (PUBYEAR, 2004) OR LIMIT-TO (PUBYEAR, 2003) OR LIMIT-TO (PUBYEAR, 2002) OR LIMIT-TO (PUBYEAR, 2001) OR LIMIT-TO (PUBYEAR, 2000) OR LIMIT-TO (PUBYEAR, 1999) OR LIMIT-TO (PUBYEAR, 1998) OR LIMIT-TO (PUBYEAR, 1997) OR LIMIT-TO (PUBYEAR, 1996)). Search date on Scopus database: 6 June 2022. Search: 334,555 document results. Quadrants: upper left (Niche Themes), upper right (Motor Themes), lower left (Emerging or Declining Themes) and lower right (Basic Themes).

Figure A4 represents the Latin American conceptual structure of research in dentistry through a thematic map of keywords during the period 1996–2020. The figure shows the quadrants of motor themes, niche themes, emerging or declining themes and basic themes.

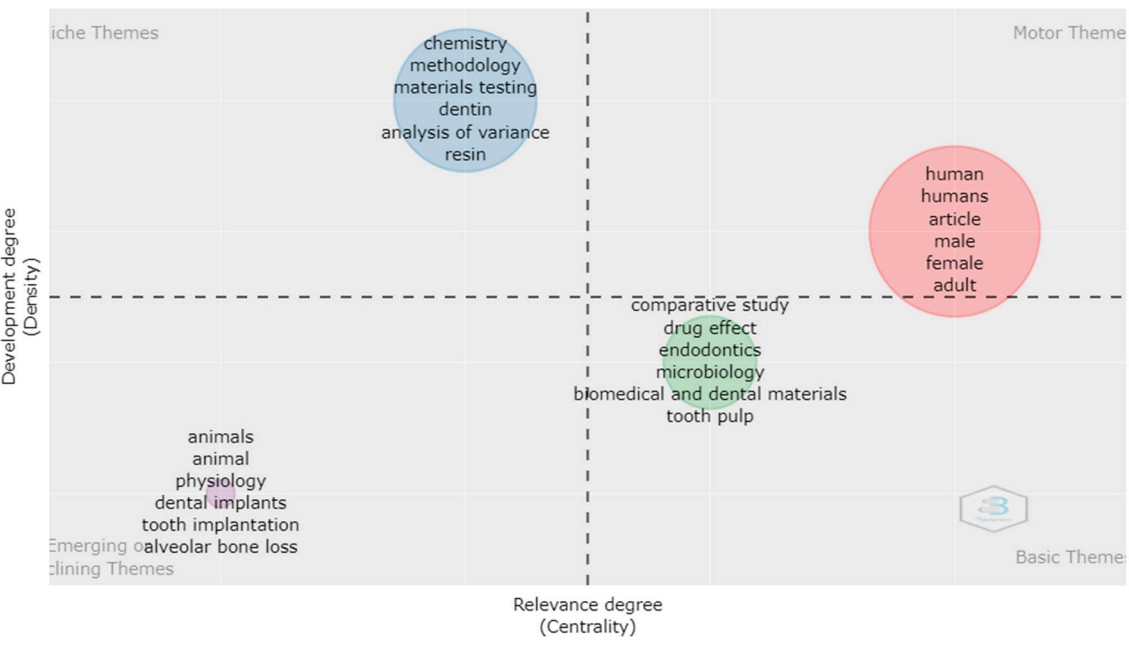

**Figure A4.** Conceptual Structure. Latin American Dentistry Thematic Map (1996–2020) generated with Bibliometrix [30]. Sort on: Cited by (highest)-First 2000 document results retrieved. Field: Keyword Plus, Number of words (560), Min cluster frequency (1–5), Number of labels (0–6), and Label size (0–0,3). Scopus Search Query: SUBJAREA (dent) AND (LIMIT-TO (AFFILCOUNTRY, "Brazil") OR LIMIT-TO (AFFILCOUNTRY, "Chile") OR LIMIT-TO (AFFILCOUNTRY, "Mexico") OR LIMIT-TO (AFFILCOUNTRY, "Colombia") OR LIMIT-TO (AFFILCOUNTRY, "Argentina") OR LIMIT-TO (AFFILCOUNTRY, "Cuba") OR LIMIT-TO (AFFILCOUNTRY, "Peru") OR LIMIT-TO (AFFILCOUNTRY, "Venezuela") OR LIMIT-TO (AFFILCOUNTRY, "Puerto Rico") OR LIMIT-TO (AFFILCOUNTRY, "Uruguay") OR LIMIT-TO (AFFILCOUNTRY, "Guatemala") OR LIMIT-TO (AFFILCOUNTRY, "Ecuador") OR LIMIT-TO (AFFILCOUNTRY, "Costa Rica") OR LIMIT-TO (AFFILCOUNTRY, "Paraguay") OR LIMIT-TO (AFFILCOUNTRY, "Trinidad and Tobago") OR LIMIT-TO (AFFILCOUNTRY, "Dominican Republic") OR LIMIT-TO (AFFILCOUNTRY, "Jamaica") OR LIMIT-TO (AFFILCOUNTRY, "Panama") OR LIMIT-TO (AFFILCOUNTRY, "Belize") OR LIMIT-TO (AFFILCOUNTRY, "Nicaragua")) AND (LIMIT-TO (PUBYEAR, 2020) OR LIMIT-TO (PUBYEAR, 2019) OR LIMIT-TO (PUBYEAR, 2018) OR LIMIT-TO (PUBYEAR, 2017) OR LIMIT-TO (PUBYEAR, 2016) OR LIMIT-TO (PUBYEAR, 2015) OR LIMIT-TO (PUBYEAR, 2014) OR LIMIT-TO (PUBYEAR, 2013) OR LIMIT-TO (PUBYEAR, 2012) OR LIMIT-TO (PUBYEAR, 2011) OR LIMIT-TO (PUBYEAR, 2010) OR LIMIT-TO (PUBYEAR, 2009) OR LIMIT-TO (PUBYEAR, 2008) OR LIMIT-TO (PUBYEAR, 2007) OR LIMIT-TO (PUBYEAR, 2006) OR LIMIT-TO (PUBYEAR, 2005) OR LIMIT-TO (PUBYEAR, 2004) OR LIMIT-TO (PUBYEAR, 2003) OR LIMIT-TO (PUBYEAR, 2002) OR LIMIT-TO (PUBYEAR, 2001) OR LIMIT-TO (PUBYEAR, 2000) OR LIMIT-TO (PUBYEAR, 1999) OR LIMIT-TO (PUBYEAR, 1998) OR LIMIT-TO (PUBYEAR, 1997) OR LIMIT-TO (PUBYEAR, 1996)). Search date on Scopus database: 6 June 2022. Search: 35,032 document results. Quadrants: upper left (Niche Themes), upper right (Motor Themes), lower left (Emerging or Declining Themes) and lower right (Basic Themes).

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
