# Peer review of "Bibliometrics Evaluation of Scientific Journals and Country Research Output of Dental Research in Latin America Using Scimago Journal and Country Rank"

_publications, doi:10.3390/publications10030026_

Round 1
Reviewer 1 Report
Title
A more specific and appropriate term for statistical evaluation... is bibliometrics evaluation.
The statement "Latin American Dentistry Investigation Ecosystem" may not be accurate to describe the scientific output analyzed. Consider to mention instead the real source of the data, Scimago Journal and Country Rank portal.
It is recommended to employ the term “scientific output” instead of “scientific production”.
Introduction
In the statement ...personal characteristics of their publications... (lines 77, 78), consider to omit the word personal.
SJR is an indicator that assesses the impact of scientific journals. It is recommended citing the concept from the original source. (Paragraph line 85)
Guerrero-Bote, V.P.; Moya-Anegón, F. A further step forward in measuring journals’ scientific prestige: The SJR2 indicator. Journal of Informetrics 2012, 6, 674–688.
Materials and Methods
Consider to delete table 1. Content could be mention in the text.
It may be convenient to add the percentage column to table 2.
Line 181 – it is mention volume of production, meanwhile the paragraph refers to the volume of citations.
Everywhere in the bibliometrics literature it is stated that citations data does not have a normal distribution. This fact could be important to discuss the indicators distributions findings.
Discussion
Brazil has the biggest output in LAC in all areas, not only in dental research. This statement may give some lights to understand the communication patron in dental research. On the other hand, the patron of scientific communication in the LAC region is similar to all areas set in productivity and citation impact.
Table 5. There is no reference to [2][3] in Puerto Rico line.
Consider to integrate tables 5 and 6 with bibliometrics data in the results section. Another option is remove the tables and to compare the findings in the discussion. Add the original source of the data.
Author Response
We thank the reviewers for the useful comments and insights that helped to improve the quality of this article. We address each of the comments point by point.
Title
A more specific and appropriate term for statistical evaluation... is bibliometrics evaluation.
The statement "Latin American Dentistry Investigation Ecosystem" may not be accurate to describe the scientific output analyzed. Consider to mention instead the real source of the data, Scimago Journal and Country Rank portal.
It is recommended to employ the term “scientific output” instead of “scientific production”.
We have changed the title of the article to “Bibliometrics Evaluation of Scientific Journals and Country Research Output of Dental Research in Latin America using SCImago Journal and Country Rank”.
Introduction
In the statement ...personal characteristics of their publications... (lines 77, 78), consider to omit the word personal.
The word has been removed.
SJR is an indicator that assesses the impact of scientific journals. It is recommended citing the concept from the original source. (Paragraph line 85)
Guerrero-Bote, V.P.; Moya-Anegón, F. A further step forward in measuring journals’ scientific prestige: The SJR2 indicator. Journal of Informetrics 2012, 6, 674–688.
The paragraph has been updated to include that the SJR is an indicator that assesses the impact of scientific journals, and the reference has been added.
Materials and Methods
Consider to delete table 1. Content could be mention in the text.
The table has been deleted and the contents are now mentioned in the text.
It may be convenient to add the percentage column to table 2.
A percentage column has been added.
Line 181 – it is mention volume of production, meanwhile the paragraph refers to the volume of citations.
We fixed the line to mention the volume of citations.
Everywhere in the bibliometrics literature it is stated that citations data does not have a normal distribution. This fact could be important to discuss the indicators distributions findings.
We have included a brief paragraph discussing the non-normality of the distribution of cites.
Discussion
Brazil has the biggest output in LAC in all areas, not only in dental research. This statement may give some lights to understand the communication patron in dental research. On the other hand, the patron of scientific communication in the LAC region is similar to all areas set in productivity and citation impact.
We have included a sentence at the beginning of the Discussion section to indicate that Brazil has the largest research output in LAC.
Table 5. There is no reference to [2][3] in Puerto Rico line.
This was a mistake, we have deleted the [2][3] in Puerto Rico line.
Consider to integrate tables 5 and 6 with bibliometrics data in the results section. Another option is remove the tables and to compare the findings in the discussion. Add the original source of the data.
We have moved Table 5 to the Results section and created a new subsection for the relationship between journals and institutions in LAC. Also, we have removed the Table 6 and discussed the findings in the discussion section citing the original data source.
Reviewer 2 Report
The authors mention that language barriers in Latin America limit access to the best scientific literature to practice Evidence-Based Dentistry. This is true, but they must also mention the limited or non-existence of financing in the universities/research center to pay for subscriptions to the most important databases. The authors expose the theory that supports their research, but do not comment on the results obtained by other researchers who have studied the scientific production in Dentistry of Latin American countries. http://www.revestomatologia.sld.cu/index.php/est/article/view/1677/438 http://www.joralres.com/index.php/JOralRes/article/view/joralres.2020.020/717
Author Response
We thank the reviewers for the useful comments and insights that helped to improve the quality of this article. We address each of the comments point by point.
The authors mention that language barriers in Latin America limit access to the best scientific literature to practice Evidence-Based Dentistry. This is true, but they must also mention the limited or non-existence of financing in the universities/research center to pay for subscriptions to the most important databases. The authors expose the theory that supports their research, but do not comment on the results obtained by other researchers who have studied the scientific production in Dentistry of Latin American countries. http://www.revestomatologia.sld.cu/index.php/est/article/view/1677/438 http://www.joralres.com/index.php/JOralRes/article/view/joralres.2020.020/717
Thank you for poiting this out. We have included more comments about other researchers that studied scientific production in Latin America and the Caribbean and included a paragraph at the end of the discussions section to comment about the language barrier and the impact on financing.
Reviewer 3 Report
The paper ‘Statistical Evaluation of Scientific Journals and Country Research Productivity in the Latin American Dentistry Investigation Ecosystem’ analyzes the density research area of the LAC region in comparison with the Other regions. Paper is clearly written and it provides accurate calculations. However, I have some concerns about the methodology and idea of the work which should be addressed prior to publication:
1) The introduction should be extended. It is unclear what hypothesis the authors test in the manuscript, and how the choosing methodology helps in that. What is the hypothesis of the work? Why may the LAC region be differ from the world in this research area? Moreover, the features of dentistry as a research field are weakly described. What is common with this discipline for all countries? What is the LAC specific? Please provide the hypothesis of the work and applicability of choosing parameters.
2) The Materials and Methods should be rewritten in a more clear way.
2.1.It is a little bit confusing when authors talk about analyzing data. On one hand the documents are analyzed (citations etc.), on the other hand they talk about the journals. Why were so many different parameters chosen? Research productivity and journal policy of national journals are quite different things.
2.2. About the tests. The results of statistical tests may vary significantly depending on the distribution function of analyzed data. Please provide distribution functions of analyzed variables (H-index, citation and etc.) for the LAC region and for other data.
It is not clear why the Anderson-Darling normality test was applied to H-index and citation data? As usual these variables have skewed distribution (Weibull or something like this). How many observations were used for Anderson-Darling normality test and Fligner-Killeen test?
In addition, the Wilcoxon rank-sum test with continuity correlation usually applied to median values. As we can see from Table 1 the difference of median and mean values is huge for the LA countries. Also please provide the explanation of choosing tests.
Moreover, there is no explanation how H-index, SJR, citation per 2 years etc. were calculated. H-index of which units were calculated (researches, countries or papers)? Is it average of all researches from the region or average by the countries?
3) The results. The authors apply three tests: Anderson-Darling normality test, the Flinger-Killen test and the Wilcoxon rank-sum test with continuity correction. What is estimated: the quality of majority or the high-class tails?
4) The number of publications correlates with the number of researchers in the country. LAC countries have high variation in territory, population, etc. For comparison of the countries publication outputs (Table 1) I would suggest adding normalized values of publications by person or by R&D funding.
5) Obviously Brasilia has the largest contribution in the results. How this bias is taking into account? Maybe it should be analyzed separately or the size of countries should be takes into account.
6) The same for Table 3. The number of citations correlates with the number of publications. It should be normalized by publications
7) I suggest that comparison with the other regions as a whole is not meaningful, because these values hardly exist in real life. This variable contains regions with very different publication outputs (e.g. North America and Africa), so the distribution is very high. I would suggest comparing different regions separately.
8) Please provide the description of Figure 2, Figure 5, Figure 6 and Figure 8.
9) I would suggest adding Table 5 before the Results or in the appendix because it contains additional information. The Discussion and Conclusion section should provide the explanation of the results, what the authors tested and why the observed difference exists? Please extend this section
10) ‘The innovations in dental sciences are potentially disruptive’. What is the meaning under ‘disruptive’? This word has different meanings
11) ‘This research aims to analyze documents, citations and journals and compare the SJR…’ Citations of which units are mentioned?
Author Response
We thank the reviewers for the useful comments and insights that helped to improve the quality of this article. We address each of the comments point by point.
The paper ‘Statistical Evaluation of Scientific Journals and Country Research Productivity in the Latin American Dentistry Investigation Ecosystem’ analyzes the density research area of the LAC region in comparison with the Other regions. Paper is clearly written and it provides accurate calculations. However, I have some concerns about the methodology and idea of the work which should be addressed prior to publication:
1) The introduction should be extended. It is unclear what hypothesis the authors test in the manuscript, and how the choosing methodology helps in that. What is the hypothesis of the work? Why may the LAC region be differ from the world in this research area? Moreover, the features of dentistry as a research field are weakly described. What is common with this discipline for all countries? What is the LAC specific? Please provide the hypothesis of the work and applicability of choosing parameters.
We have explicitly included the hypothesis in the introduction section. Furthermore, we have expanded the introduction section to clarify the statement about potential disruptive innovation. We also included a paragraph stating the low priority of dental care in LAC and how it might discourage dental research in the region.
2) The Materials and Methods should be rewritten in a more clear way.
2.1.It is a little bit confusing when authors talk about analyzing data. On one hand the documents are analyzed (citations etc.), on the other hand they talk about the journals. Why were so many different parameters chosen? Research productivity and journal policy of national journals are quite different things.
This research aimed to analyse both documents, citations and journals to assess the dental research output of Latin America and the Caribbean in contrast to other regions. To do so, we decided to include information about the countries and the institutions related to the journals. We consider that this information, additional to the journal indicators, is useful to provide a clearer picture of the current state of dental research in the region.
2.2. About the tests. The results of statistical tests may vary significantly depending on the distribution function of analyzed data. Please provide distribution functions of analyzed variables (H-index, citation and etc.) for the LAC region and for other data.
We have included Cullen and Frey graphs to evaluate the distribution functions of the variables.
It is not clear why the Anderson-Darling normality test was applied to H-index and citation data? As usual these variables have skewed distribution (Weibull or something like this). How many observations were used for Anderson-Darling normality test and Fligner-Killeen test?
The Anderson-Darling test is a powerful statistical tool and can be used to detect departures from the normal distribution. It is already known that, most of the time, the H-Index, SJR and citations metrics don’t follow a normal distribution; therefore, our intention was to ensure that the measurements obtained for this research were also not normally distributed. Also, we have added a paragraph to the results section to indicate that there were 201 journals included in this study, from which 12 correspondend to the LAC region.
In addition, the Wilcoxon rank-sum test with continuity correlation usually applied to median values. As we can see from Table 1 the difference of median and mean values is huge for the LA countries. Also please provide the explanation of choosing tests.
We have included a statatement that the Fligner-Killeen test of homogeneity of variances proved that the variances of the H-Index of LAC and Other Regions are not equal, therefore we can only test for differences in the mean of the two groups. We could not test for difference of median values.
Moreover, there is no explanation how H-index, SJR, citation per 2 years etc. were calculated. H-index of which units were calculated (researches, countries or papers)? Is it average of all researches from the region or average by the countries?
The H-Index, SJR and citation metrics were calculated and provided by SCImago Journal and Country Rank.
3) The results. The authors apply three tests: Anderson-Darling normality test, the Flinger-Killen test and the Wilcoxon rank-sum test with continuity correction. What is estimated: the quality of majority or the high-class tails?
In this research we estimated the differences in the distributions of two samples, and where the majority of the observations were located.
4) The number of publications correlates with the number of researchers in the country. LAC countries have high variation in territory, population, etc. For comparison of the countries publication outputs (Table 1) I would suggest adding normalized values of publications by person or by R&D funding.
Thank you for pointing this out. We have used one normalised metric (citations per document) to lessen the impact of population size or number of publications. This provided interesting results, as there were no significant differences in the number of cites per document between LAC and Other regions.
5) Obviously Brasilia has the largest contribution in the results. How this bias is taking into account? Maybe it should be analyzed separately or the size of countries should be takes into account.
Brazil has the largest contribution to scientific research in the region; but specially in dental research, as Brazil is the dental industry powerhouse of the region. There is a need to further explore the impact of Brazil in dental research, however it is beyond the scope of this article.
6) The same for Table 3. The number of citations correlates with the number of publications. It should be normalized by publications
This is covered by the cites per document metric.
7) I suggest that comparison with the other regions as a whole is not meaningful, because these values hardly exist in real life. This variable contains regions with very different publication outputs (e.g. North America and Africa), so the distribution is very high. I would suggest comparing different regions separately.
In this article we wanted to focus on the research output of the LAC region. A comparison between LAC and regions such as North America or Europe would require an in-depth analysis of those other regions as well. In this regard, it would be interesting to perform a comparison between the research output of LAC and North America, as there are direct contact points between these two regions.
8) Please provide the description of Figure 2, Figure 5, Figure 6 and Figure 8.
Thank you for pointing this out. The previous changes now provide more description to the figures.
9) I would suggest adding Table 5 before the Results or in the appendix because it contains additional information. The Discussion and Conclusion section should provide the explanation of the results, what the authors tested and why the observed difference exists? Please extend this section
Theinformation of Table 5 has been moved up to the Results section following the suggestions of another reviewer.
10) ‘The innovations in dental sciences are potentially disruptive’. What is the meaning under ‘disruptive’? This word has different meanings
In this case, we refer to new developments that can permanently change the way that dental care is provided.
11) ‘This research aims to analyze documents, citations and journals and compare the SJR…’ Citations of which units are mentioned?
In this case we refer to the total cites of the last 3 years and the cites per document of the last 2 years.
Reviewer 4 Report
The study has important virtues. The first is that it fills a knowledge gap regarding the current state of dental research in Latin America. Secondly, the paper is sufficiently well articulated, it uses a large and well-maintained database such as Scimagojr, and the methodology is fine. The paper does not, however, participate in the current debate on the importance of scientific journals in Latin America and the countries of the global south. Precisely, the problem with the use of metrics such as the number of citations, the H index and the SJR is that they do not inform about the relevance of the knowledge produced. On the contrary, historically in Latin America the use of these metrics has been abused as tools for the promotion of scientific careers and decision-making regarding research funding. In this sense, I strongly recommend the authors to read and include in the discussion the Leiden Manifesto for research metrics:
Hicks, D., Wouters, P., Waltman, L., De Rijcke, S., & Rafols, I. (2015). Bibliometrics: the Leiden Manifesto for research metrics. Nature, 520(7548), 429-431.
As has been reported in the literature in various fields of knowledge, Latin America is expected to have modest bibliometric results compared to Europe or the United States.
Meneghini, R., Mugnaini, R., & Packer, A. L. (2006). International versus national oriented Brazilian scientific journals. A scientometric analysis based on SciELO and JCR-ISI databases. Scientometrics, 69(3), 529-538.
Meneghini, R., Mugnaini, R., & Packer, A. L. (2006). International versus national oriented Brazilian scientific journals. A scientometric analysis based on SciELO and JCR-ISI databases. Scientometrics, 69(3), 529-538.
However, this fact, already expected, gives little information about the relevance of research in the subcontinent. Even more little serves as a guide for research policies. Should journals in the region focus on the European or US market in order to improve their numbers?
In order for the article to be truly relevant and participate in the current bibliometric and scientometric debates, a qualitative analysis of the content of Latin American dental research journals should be carried out versus other regions. It is easy! There are already several freely accessible and easy-to-use tools at the disposal of authors.
Vosviewer generates, for example, keyword concurrency maps that can be used to generate hypotheses about what is happening with dental research in each region. The KH coder tool can be used to statistically analyze content and test hypotheses. For example, it would be essential to understand whether dental research in Latin America meets the health needs of the region or whether it plays a peripheral role with respect to the global North.
I recommend the following article as a source of methodological ideas:
Fytilakos, I. (2021). Text mining in fisheries scientific literature: A term coding approach. Ecological Informatics, 61, 101203.
Author Response
We thank the reviewers for the useful comments and insights that helped to improve the quality of this article. We address each of the comments point by point.
The study has important virtues. The first is that it fills a knowledge gap regarding the current state of dental research in Latin America. Secondly, the paper is sufficiently well articulated, it uses a large and well-maintained database such as Scimagojr, and the methodology is fine. The paper does not, however, participate in the current debate on the importance of scientific journals in Latin America and the countries of the global south. Precisely, the problem with the use of metrics such as the number of citations, the H index and the SJR is that they do not inform about the relevance of the knowledge produced. On the contrary, historically in Latin America the use of these metrics has been abused as tools for the promotion of scientific careers and decision-making regarding research funding. In this sense, I strongly recommend the authors to read and include in the discussion the Leiden Manifesto for research metrics:
Hicks, D., Wouters, P., Waltman, L., De Rijcke, S., & Rafols, I. (2015). Bibliometrics: the Leiden Manifesto for research metrics. Nature, 520(7548), 429-431.
As has been reported in the literature in various fields of knowledge, Latin America is expected to have modest bibliometric results compared to Europe or the United States.
Meneghini, R., Mugnaini, R., & Packer, A. L. (2006). International versus national oriented Brazilian scientific journals. A scientometric analysis based on SciELO and JCR-ISI databases. Scientometrics, 69(3), 529-538.
We would like to thank you for your comments and suggestions. We have included a paragraph in the discussions section about the implications of the third principle of the Leiden Manifesto for dental research in Latin America and the Caribbean. Furthermore, we have made several changes to the manuscript to include a qualitative analysis of Latin American dental research.
However, this fact, already expected, gives little information about the relevance of research in the subcontinent. Even more little serves as a guide for research policies. Should journals in the region focus on the European or US market in order to improve their numbers?
The impact of metrics on the funding of scientific research in LAC is a very interesting topic that needs in depth analysis. Personally, we consider that journals from LAC should focus in themes that are relevant to the region, even if that slow downs the growth in metrics.
In order for the article to be truly relevant and participate in the current bibliometric and scientometric debates, a qualitative analysis of the content of Latin American dental research journals should be carried out versus other regions. It is easy! There are already several freely accessible and easy-to-use tools at the disposal of authors.
Vosviewer generates, for example, keyword concurrency maps that can be used to generate hypotheses about what is happening with dental research in each region. The KH coder tool can be used to statistically analyze content and test hypotheses. For example, it would be essential to understand whether dental research in Latin America meets the health needs of the region or whether it plays a peripheral role with respect to the global North.
I recommend the following article as a source of methodological ideas:
Fytilakos, I. (2021). Text mining in fisheries scientific literature: A term coding approach. Ecological Informatics, 61, 101203.
We included a Bibliometrix analysis charts in the Appendix A section, this is an alternative to Vosviewer that is also broadly used and provides some additional functionality such as thematic maps. Thank you for pointing this out.
Round 2
Reviewer 3 Report
I appreciate a lot the works that the authors provide in favor to improve the manuscript but I still have several crucial comments and suggestions.
General comments:
The manuscript contains many variables and calculations which now do not joined by one message. The authors analyze the number of publications and its citation, journal characteristics and graphs of keywords. What these variables and its values said about dentistry discipline in LAC region? Which conclusion the readers should make from the work? Please provide clear message of the work.
The authors said that LAC region is studied. Actually, only one big country is studied – Brazil (82, 94% of LAC publications) because the authors do not take into account the size of the countries.
The manuscript contains many descriptive results without explanation. For example, ‘Table 4 provides a summary of the relationships between the dentistry journals and 306 research institutions in LAC, alongside the ARWU 2020 and QS 2020 metrics in the case 307 of Universities. The results show that Brazil accumulates top three dentistry journals in 308 LAC are related to Brazilian universities ranked in the top 101-150 positions of the ARWU 309 2020 [33] and the top 116 of the QS 2020 [34] indices.’ How positions of publisher relate with dentistry of LAC region?
1) There is some misunderstanding in introduction. The authors describe the features of LAC region that distinguish this region from others. At the same time, the hypothesis is formulated as ‘no difference’ with other region. On which facts the hypothesis is based?
2) In section ‘2. Materials and Methods’ there is no description of Cullen and Frey graphs and World Dentistry Co-occurrence Network although these techniques use in the manuscript. Also please provide the information how the SJR was calculated. Moreover, in introduction there is no explanation for which purpose provides network analysis. Which type of network was used? How we can interpret the results? Now part with the network gives more questions than explanation. It should be expanded or deleted.
3) Table 1 and Table 2 provide the share of publications and citation by countries. But this does not give any additional information about the countries and does not make the countries comparable. From these values we see that Brazil is the lagers player, but we know nothing about productivity of the countries in this research area. Moreover, I do not think that this is correct to calculate the share of citation by countries in joint publication output. For example, if Brazil and Chili have join publication. When this publication was cited, who receive the score? If both, the sum of countries shares should be more 100%
4) P.8 ‘The Cullen and Frey graph (Figure 4) shows that the distribution of the SJR of 243 Journals of the LAC region can be explained by a beta distribution…..On the 245 other hand, this graph suggests that the H-Index of Journals from Other regions is close 246 to a lognormal distribution…’
Is it about SJR or H-index or its comparison?
4) In conclusion there is no explanation about the rejection or acceptance of postulated hypothesis. Can the hypothesis were rejected or not and why?
Author Response
I appreciate a lot the works that the authors provide in favour to improve the manuscript, but I still have several crucial comments and suggestions.
General comments:
The manuscript contains many variables and calculations which now do not joined by one message. The authors analyse the number of publications and its citation, journal characteristics and graphs of keywords. What these variables and its values said about dentistry discipline in LAC region? Which conclusion the readers should make from the work? Please provide clear message of the work.
Thank you for pointing this out. We have included a paragraph at the beginning of the conclusions section that indicates the core findings, and their relationships with the variables used.
The authors said that LAC region is studied. Actually, only one big country is studied – Brazil (82, 94% of LAC publications) because the authors do not take into account the size of the countries.
This study took into account the number of cites per document, that is a normalized metric for journal quality assessment, in relationship with the size of the journal. However, normalizing the metrics by the population of the countries would require another full set of analysis to evaluate the internal differences between the production of countries. This is a very interesting topic, but would also require focusing on regional journals and not only those indexed in Scopus. We have included this as a future research line.
The manuscript contains many descriptive results without explanation. For example, ‘Table 4 provides a summary of the relationships between the dentistry journals and 306 research institutions in LAC, alongside the ARWU 2020 and QS 2020 metrics in the case 307 of Universities. The results show that Brazil accumulates top three dentistry journals in 308 LAC are related to Brazilian universities ranked in the top 101-150 positions of the ARWU 309 2020 [33] and the top 116 of the QS 2020 [34] indices.’ How positions of publisher relate with dentistry of LAC region?
The ranking of the publisher, as academic institutions, provides insights about the quality of the integration of academic activities and scientific research. This has now been clearly indicated in the article.
1) There is some misunderstanding in introduction. The authors describe the features of LAC region that distinguish this region from others. At the same time, the hypothesis is formulated as ‘no difference’ with other region. On which facts the hypothesis is based?
These are the null hypothesis that were tested. This has been corrected in the introduction section.
2) In section ‘2. Materials and Methods’ there is no description of Cullen and Frey graphs and World Dentistry Co-occurrence Network, although these techniques use in the manuscript.
The Cullen and Frey graph and the Co-occurrence Network methods are now described in the Materials and Methods section.
Also please provide the information how the SJR was calculated.
The SJR is a metric provided by SCImago Journal and Country Rank and it was not calculated by the researchers.
Moreover, in introduction, there is no explanation for which purpose provides network analysis. Which type of network was used? How we can interpret the results? Now part with the network gives more questions than explanation. It should be expanded or deleted.
The co-occurrence network method was used to find the clusters of keywords most frequently used in dental research. The results of this network are now discussed in the discussion section and provide guidelines to the conclusions.
3) Table 1 and Table 2 provide the share of publications and citation by countries. But this does not give any additional information about the countries and does not make the countries comparable. From these values, we see that Brazil is the lagers' player, but we know nothing about productivity of the countries in this research area. Moreover, I do not think that this is correct to calculate the share of citation by countries in joint publication output. For example, if Brazil and Chili have joined publication. When this publication was cited, who receive the score? If both, the sum of countries shares should be more 100%
These metrics are provided by SCImago Journal and Country Rank and take into account joint publication output, therefore the percentage is calculated after allocating the number of documents and citations.
4) P.8 ‘The Cullen and Frey graph (Figure 4) shows that the distribution of the SJR of 243 Journals of the LAC region can be explained by a beta distribution…..On the 245 other hand, this graph suggests that the H-Index of Journals from Other regions is close 246 to a lognormal distribution…’
Is it about SJR or H-index or its comparison?
Thank you for pointing this out. It is about the SJR.
4) In conclusion, there is no explanation about the rejection or acceptance of postulated hypothesis. Can the hypothesis were rejected or not and why?
The explanation about the rejection or acceptance of the postulated hypothesis is now detailed in the conclusion section.
Reviewer 4 Report
The authors have considerably improved the manuscript. There are still some improvements to be made: In the co-occurrence networks, there are repeated or very general terms that do not contribute to interpretation. This can be corrected with the use of thesaurus and stop worlds. The visualization of the maps should be improved avoiding the overlap of the text.
More importantly, the authors should incorporate a brief analysis into the discussion of whether based on the maps that the authors elaborated (figures A1 to A4), the academic dental literature of Latin America differs from the rest of the world.
Author Response
The authors have considerably improved the manuscript. There are still some improvements to be made: In the co-occurrence networks, there are repeated or very general terms that do not contribute to interpretation. This can be corrected with the use of thesaurus and stop worlds. The visualization of the maps should be improved avoiding the overlap of the text.
More importantly, the authors should incorporate a brief analysis into the discussion of whether based on the maps that the authors elaborated (figures A1 to A4), the academic dental literature of Latin America differs from the rest of the world.
Thank you for pointing this out. In the case of synonyms and stopwords, we tried to prevent the inclusion of biases by our researchers. In another scenario, with a different scope, we would have proposed a protocol to verify that the synonyms and stopwords are valid and accepted by a panel of experts and previous literature. Nevetheless, the current output of the Co-ocurrence Network provides good information about the trends in dental research for both the LAC region and the world, and these findings proved useful to assess the possible causes behind the lower impact metrics of the LAC region. Furthermore, we included a brief explanation regarding this in the discussions section.
Round 3
Reviewer 3 Report
The authors do a great job improving the manuscript. With that, there are still several comments that should be addressed prior publication.
1) I agree with using the journal normalize metrics, it is comparable. As we see from Table 1, LAC dentistry research is concentrated in one country – Brazil. There is the problem with differences in publication output of the countries. The authors say that they analyze LA countries, but in fact they mostly analyze Brazil since 82.9% of analyzed publications are from Brazil. Therefore, the observed results are relevant for Brazil and cannot be relevant for the other countries. In conclusion, authors should indicate this limitation.
“This study was centered around journals indexed by Scopus, where most of the journals of the region originated from Brazil..” I do not think that using another database will solve the problem. To do it, the LA countries should be studied separately or its publication output should be normalized.
2) There is lack information on network analysis in the introduction and abstract. I think it should be mentioned why this technique was used.
3) Research questions (RQ) in introduction repeat the hypothesis. In my previous review I recommended to add hypothesis to perform rigor statistical analysis with certain significance level. At the same time, RQ also can be presented prior to the hypothesis to show the goal of the work. As usual, research questions are formulated wider than hypotheses. Hypotheses help to answer research questions. For example, a research question is to analyze the difference in publication output between two countries. The hypothesis (that can be statistically proven) is: the number of publications of two countries is equal.
4) P.16 “On the other hand, previous research suggested….” The references are needed
Author Response
The authors do a great job improving the manuscript. With that, there are still several comments that should be addressed prior publication.
1) I agree with using the journal normalize metrics, it is comparable. As we see from Table 1, LAC dentistry research is concentrated in one country – Brazil. There is the problem with differences in publication output of the countries. The authors say that they analyze LA countries, but in fact they mostly analyze Brazil since 82.9% of analyzed publications are from Brazil. Therefore, the observed results are relevant for Brazil and cannot be relevant for the other countries. In conclusion, authors should indicate this limitation.
“This study was centered around journals indexed by Scopus, where most of the journals of the region originated from Brazil..” I do not think that using another database will solve the problem. To do it, the LA countries should be studied separately or its publication output should be normalized.
Thank you pointing this out, we have added this at the end of the future research lines.
2) There is lack information on network analysis in the introduction and abstract. I think it should be mentioned why this technique was used.
We have added a brief description in the introduction section and mentioned it in the abstract.
3) Research questions (RQ) in introduction repeat the hypothesis. In my previous review I recommended to add hypothesis to perform rigor statistical analysis with certain significance level. At the same time, RQ also can be presented prior to the hypothesis to show the goal of the work. As usual, research questions are formulated wider than hypotheses. Hypotheses help to answer research questions. For example, a research question is to analyze the difference in publication output between two countries. The hypothesis (that can be statistically proven) is: the number of publications of two countries is equal.
We have changed RQ2 and RQ3 to better explain the aim of the study, and to include the analysis of keywords using network analysis.
4) P.16 “On the other hand, previous research suggested….” The references are needed
Thank you for pointing this out. We have included the corresponding references to this section.
